# 3DSMT: A Hybrid Spiking Mamba-Transformer for Point Cloud Analysis

**Zhiming Zhou**[1], **Yong He**[1*], **Chaoxu Mu**[1,2], **Qiaoyun Wu**[3], **Ajmal Mian**[4]
[1]Anhui University
[2]Tianjin University
[3]Nanjing University of Aeronautics and Astronautics
[4]The University of Western Australia
wa23301159@stu.ahu.edu.cn, h.yong@hnu.edu.cn
cxmu@tju.edu.cn, wuqiaoyun@nuaa.edu.cn, ajmal.mian@uwa.edu.au

## ABSTRACT

The sparse unordered structure of point clouds causes unnecessary computation and energy consumption in deep models. Conventionally, the Transformer architecture is leveraged to model global relationships in point clouds, however, its quadratic complexity restricts scalability. Although the Mamba architecture enables efficient global modeling with linear complexity, it lacks natural adaptability to unordered point clouds. Spiking Neural Network (SNN) is an energy-efficient alternative to Artificial Neural Network (ANN), offering an ultra low-power event-driven paradigm. The inherent sparsity and event-driven characteristics of SNN are highly compatible with the sparse distribution of point clouds. To balance efficiency and performance, we propose a hybrid spiking Mamba-Transformer (3DSMT) model for point cloud analysis. 3DSMT integrates a Spiking Local Offset Attention module to efficiently capture fine-grained local geometric features with a spiking Mamba block designed for unordered point clouds to achieve global feature integration with linear complexity. Experiments show that 3DSMT achieves state-of-the-art performance among SNN-based methods in shape classification, few-shot classification, and segmentation tasks, significantly reducing computational energy consumption while also outperforming numerous ANN-based models. *Our code is available.*

## 1 INTRODUCTION

Point cloud analysis is essential for enabling machines to perceive and understand complex 3D environments, driving a wide range of applications such as autonomous driving, robotics, augmented/virtual reality, and metaverse interactions. While significant progress has been made in point cloud processing using Artificial Neural Network (ANN) Priddy (2005), achieving high accuracy often comes at the cost of substantial computational complexity and energy consumption. Energy inefficiency poses a major barrier to deploying advanced 3D perception systems on resource-constrained edge devices, such as drones, mobile robots, and AR/VR headsets, which require real-time processing and have limited power. Consequently, the development of high-performance yet energy-efficient models for point cloud analysis is emerging as a critical and formidable research frontier, essential for advancing practical applications.

Transformer architectures, particularly variants such as Point Transformer Zhao et al. (2021), have demonstrated remarkable success in modeling complex relationships in point clouds, further advancing the capabilities of ANN-based methods in this domain. Their self-attention mechanism excels at capturing global dependencies, but the standard dot product operation used in these architectures has quadratic computational complexity ($\mathcal{O}(N^2)$) with respect to the number of points ($N$). This high computational cost severely limits its applicability to large-scale point clouds, despite its effectiveness in global modeling. To address this challenge, localized attention mechanisms have been adopted to reduce energy consumption and improve efficiency in point cloud processing.

---

*Corresponding author: Yong He (h.yong@hnu.edu.cn)

Recently, state-space model (SSM) such as Mamba Gu & Dao (2023) have emerged as an alternative solution offering linear computational complexity ($\mathcal{O}(N)$) and enabling efficient global context integration. Compared to Transformers, Mamba consumes less energy and yet captures long-range dependencies across large scale point clouds.

However, even with these advancements, the computational and energy demands of point cloud analysis remain high, especially for real-time applications on resource-constrained devises. To further address these challenges, Spiking Neural Network Maass (1997), inspired by the brain's information processing, have emerged as a disruptive paradigm, already demonstrating significant progress in 2D image analysis Zhou et al. (2023b;a). SNN communicate via sparse, event-driven binary spikes and leverage energy-efficient addition operations to achieve ultra-low computational energy consumption on neuromorphic hardware Roy et al. (2019). Their event-driven nature aligns perfectly with the sparse distribution of point clouds, while their temporal dynamics provide a natural framework for sequential feature reasoning, making them highly promising for sustainable 3D perception.

To combine the energy efficiency of SNN with powerful local and global modeling capabilities of Transformer and Mamba, we propose a hybrid spiking Mamba-Transformer for point cloud analysis (coined **3DSMT**). The core insight is to jointly integrate attention-based local modeling and SSM-based global modeling within an spiking framework. Our SNN-based 3DSMT model is applicable to a wide range of point cloud analysis tasks. Our contributions are summarized as follows:

- We propose Spiking Local Offset Attention that captures fine-grained geometric structures through sparse, event-driven computation, reducing computational and energy costs.

- We introduce a novel Spiking Mamba Block that achieves efficient global feature integration with linear complexity and low energy consumption, leveraging Mamba's strengths in processing large-scale sequences.

- We present 3DSMT, a hybrid spiking architecture that combines the local feature extraction of Spiking Local Offset Attention and the global feature integration of the Spiking Mamba Block, within an energy-efficient spiking paradigm, for point clouds analysis.

Our method achieves state-of-the-art performance of 92.0%, 92.1% and 90.6% Overall Accuracy (OA) on the three variants (i.e., PB_T50_RS, OBJ_BG, OBJ_ONLY) of ScanObjectNN Uy et al. (2019). Additionally, it achieved 95.2% OA on ModelNet40 Wu et al. (2015) for classification with reduced energy consumption. In the few-shot classification setting on ModelNet40, our method achieves 92.8% and 96.2% OA in 5-way 10-shot and 20-shot settings, respectively, and 87.2% and 92.1% in 10-way 10-shot and 20-shot settings. Furthermore, it achieves 85.1% instance mean Intersection over Union (Ins.mIoU) on ShapeNetPart Yi et al. (2016) for part segmentation which is the highest among SNN-based methods. Notably, our method also outperforms the majority of ANN-based methods while significantly reducing computation and energy consumption.

## 2  RELATED WORK

**ANN-based Point Cloud Analysis.** Deep neural network architectures for point cloud data based on traditional ANN can be roughly divided into projection-based, voxel-based, and point-based methods. Projection-based methods simplify 3D tasks by converting them into 2D image problems, often leading to information loss. Voxel-based methods transform point clouds into voxel grids and apply 3D convolutions, but the computational cost grows cubically with resolution. PointNet Qi et al. (2017a), a pioneering point-based method, uses MLP to extract global information from unordered point sets. PointNet++ Qi et al. (2017b) further introduces hierarchical feature learning to capture local geometric structures, since becoming the cornerstone of modern point cloud analysis.

With the development of deep learning, point cloud analysis has shifted from MLP-based methods to transformer-based approaches. Point Transformer Zhao et al. (2021), which fuses local and global features, has become the state-of-the-art backbone for point cloud analysis. However, its quadratic complexity increases computational costs, posing challenges for long-sequence point clouds. Recently, Mamba with linear complexity and powerful modeling for capturing long-range dependencies—has emerged as a more efficient alternative. PointMamba Liang et al. (2024) first introduces the Mamba architecture into point cloud processing, significantly improving efficiency while maintaining high accuracy. Nevertheless, both methods are constrained by the inherent energy

inefficiency of traditional artificial neural network, which remains an urgent problem to solve in the field of point cloud analysis.

**SNN-based Point Cloud Analysis.** The integration of spiking neural network with point cloud processing is a novel and energy-efficient approach, but remains in its early stages of development. Spiking PointNet Ren et al. (2024) proposes a 'less training but more learning' framework based on PointNet, being the first to apply SNN to point clouds. P2SResLnet Wu et al. (2024a) combines spiking neurons with traditional point convolutions to construct a point-to-spike residual classification network. SPT Wu et al. (2025a) designs a queue-driven sampling direct encoding method for point clouds, building the first Transformer-based point cloud classification framework. SPM Wu et al. (2025b) is the first spike point cloud analysis framework based on Spike Mamba. It not only brings spike-based pre-training into the point cloud field, but also uses a two-branch time-flipping strategy to fit spike point cloud processing. However, our proposed 3DSMT effectively integrates Mamba with point clouds through a strategy that combines two-branch horizontal flipping and channel flipping. Current, the above SNN models still have a significant accuracy performance gap compared with ANN models, and further improving the accuracy of SNN based point cloud analysis models remains an open challenge.

Current research on integrating SNN with point cloud data faces significant accuracy gaps. The core issues lie in insufficient feature representation capabilities, immature training mechanisms, and a trade-off that often sacrifices accuracy excessively for lower energy consumption. To bridge this gap and fully harness the potential of spiking neural network in point cloud processing, it is crucial to develop an architecture that balances accuracy and energy efficiency. Our proposed 3DSMT addresses these challenges by leveraging Integrate-and-Fire (IF) neurons Bulsara et al. (1996), which excel in energy-sensitive tasks due to their low memory and energy requirements. By adhering strictly to direct training with surrogate gradients, 3DSMT avoids the additional temporal overhead associated with ANN-SNN conversion, ensuring low energy consumption. Furthermore, 3DSMT effectively combines the high-performance characteristics of Transformer and Mamba models in point cloud processing, achieving an optimal energy-accuracy balance.

## 3 METHOD

We propose hybrid spiking Mamba-Transformer network (3DSMT) that delivers high performance at low energy consumption. 3DSMT is a spiking neural network specifically designed for point cloud analysis. Below, we outline the overall 3DSMT framework, followed by a comprehensive exploration of its critical design components.

### 3.1 3DSMT OVERVIEW

As illustrated in the top of Figure 1, the 3DSMT architecture composes three key components: Spiking Patch Embedding, Spiking Hybrid Block and Prediction Head. The following sections will delve into the details of each components.

**Spiking Patch Embedding.** Given an input point cloud $P \in \mathbb{R}^{N \times 3}$ containing $N$ points, we first use the farthest point sampling (FPS) to select $L$ center points $P_{ct} \in \mathbb{R}^{L \times 3}$. Then, for each center point $P_{ct}^i$, we construct a local patch $\mathbf{x}_p^i \in \mathbb{R}^{k \times 3}$ using K-nearest neighborhood (KNN). Note that spiking neurons endow the network with spatio-temporal characteristics, and $T$ is a key hyperparameter for time delay. $\mathbf{x}_p^i$ is copied for each time step $t \in [0, T)$ as input $\mathbf{E} \in \mathbb{R}^{T \times k \times 3}$ to the Spiking Patch Embedding (SPE). As shown in Figure 1, SPE can be formulated as,

$$U = \mathrm{MLP}(\mathrm{SN}(\mathrm{MLP}(E))), \quad G = \mathrm{MaxPool}(U), \quad U' = \mathrm{MLP}(\mathrm{SN}(\mathrm{Concat}(U, G))), \quad (1,2,3)$$

here the membrane potential $U$ is the preliminarily learned deeper semantic information, $G$ is the output of MaxPool on $U$ to capture the local context of each token, and $U'$ is the output obtained by processing the concatenation of $U$ and $G$ through the SN layer and MLP. The MLP in this context is implemented using a 1D convolution layer (Conv1D) and a Batch Normalization layer (BN). The SN layer, on the other hand, consit of a Spiking Neuron Layer.

**Spiking Hybrid Block.** The acquired patch embeddings are regarded as token sequences in Transformer architecture, analogous to ViT Dosovitskiy et al. (2020). Specifically, we introduce a learnable [*CLS*] token to aggregate information of the entire sequence. Each patch $x_p \in \mathbb{R}^{L \times k \times 3}$ is

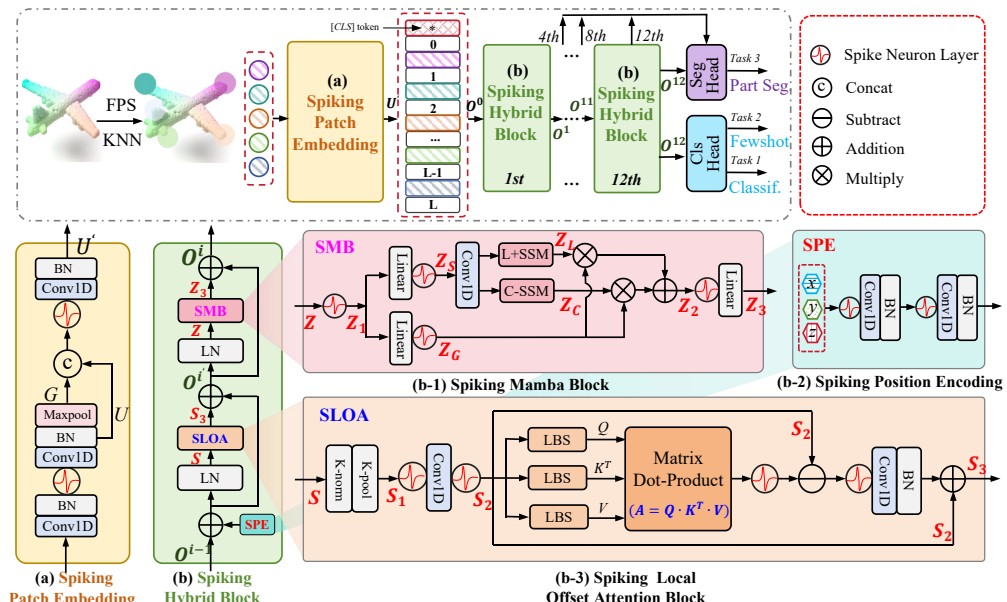

Figure 1: 3DSMT overview. The model comprises a Spiking Patch Embedding (SPE) module, a sequence of Spiking Hybrid Blocks (SHBs), and a task-specific head. The output of the (i-1)-th SHB serves as the input to the i-th SHB. (a) The SPE module first maps low-dimensional point coordinates into a high-dimensional feature space, which serves as the input to the first SHB. (b) Each SHB integrates a Spiking Local Offset Attention (SLOA) block, a Spiking Mamba Block (SMB), and a Spiking Position Encoding (SPE) module to capture local and global features.

projected into a $C$-dimensional feature vector $F_p(x_p) \in \mathbb{R}^{L \times C}$, serving as its initial patch embedding. After incorporating position encoding, the [CLS] token is vertically stacked with all point cloud patch features to form a geometric structure information sequence of shape $(L+1) \times C$ (consisting of $L$ patch features plus 1 learnable [CLS] token). This sequence is subsequently fed into the Spiking Hybrid Block (SHB) for high-level feature learning. As illustrated at the bottom of Figure 1. SHB comprises Spiking Local Offset Attention (SLOA) and Spiking Mamba Block (SMB), which are used to extract local and global features, respectively. Each module is preceded by a Layer Norm layer (LN) and employs residual connections. Specifically, it can be described as,

$$O^0 = [X_{cls}; F_p(x_p^1); \cdots; F_p(x_p^L)] + S_{pos}, \tag{4}$$

$$O^{i'} = \text{SLOA}(\text{LN}(O^{i-1} + S_{pos})) + O^{i-1}, \tag{5}$$

$$O^i = \text{SMB}(\text{LN}(O^{i'})) + O^{i'}, i = 1 \ldots M, \tag{6}$$

where $O^i$ denotes the output of each SHB layer, $X_{cls} \in \mathbb{R}^{1 \times C}$ is the learnable [CLS] token, and LN signifies the LayerNorm operation. In practice, we stack $M = 12$ SHBs, and each SHB incorporates Spiking Position Encoding $S_{pos} \in \mathbb{R}^{(L+1) \times C}$ to enhance the model's spatial awareness.

**Prediction Head.** The prediction head includes a classification head and a segmentation head. *Classification Head:* Stacked linear layers that map high-level SHB features to class predictions. It excels in both standard shape classification and few-shot learning with limited labels. *Segmentation Head:* Inspired by PointBERT Yu et al. (2022), it combines features from the 4th, 8th, and 12th SHB layers (like PointNet++) and outputs point-wise part probabilities for fine-grained segmentation.

### 3.2 SPIKING LOCAL OFFSET ATTENTION

Local geometric features are crucial for point cloud feature learning. To exploit these features, Mamba3D Han et al. (2024) use K-Norm to propagate local features from the central point to adjacent points, while K-Pool effectively performs local feature aggregation. Inspired by this, we propose a Spiking Local Offset Attention (SLOA). As shown in Figure 1, given the input $S$, we first use K-Norm and K-Pool for local feature propagation and aggregation to obtain $S_1$. Then, the SN

layer converts the local features into a spiking sequence {0,1} with spatio-temporal information, followed by channel alignment through a MLP. The features are then converted into a spiking sequence $S_2$ via the SN layer again. Finally, we obtain query $Q$, key $K$, and value $V$ matrics through linear transformations and feed them into distinct spiking neurons. We abbreviate the sequential operation of Linear, Batch Normalization, and Spiking Layer as LBS, which can be expressed as,

$$S_1 = \text{K-Pool(K-Norm}(S)), \quad S_2 = \text{SN(MLP(SN}(S_1))), \tag{7,8}$$

$$Q = \text{LBS}(S_2), \quad K = \text{LBS}(S_2), \tag{9,10}$$

$$V = \text{LBS}(S_2), \quad A = Q \cdot K^T \cdot V, \tag{11,12}$$

where $Q$, $K$, $V \in \mathbb{R}^{T \times N \times D}$ are represented in spiking form. Due to the inherent sparsity and non-negativity of spiking sequences, the calculation of the attention matrix $A$ can be directly accomplished using simple logical AND operations and accumulation (AC) operations. This avoids the energy-intensive floating-point multiply-accumulate (MAC) operations caused by the softmax function in traditional ANN, dramatically reducing computational energy consumption. Subsequently, the attention layer computes the offset between the attention features and the input feature $S_2$ through element-wise subtraction. After processing the offset through the SN and MLP, it is added to the input feature $S_2$ to obtain the output feature $S_3$, expressed as,

$$S_3 = \text{MLP(SN}(S_2 - \text{SN}(A))) + S_2. \tag{13}$$

By introducing this spiking offset attention mechanism, SLOA can more explicitly highlight the differences in features within local regions of point clouds, strengthen the ability to capture subtle structural changes, effectively improve the fine-grained local features representation and overall analysis performance.

### 3.3 SPIKING MAMBA BLOCK

ViM Zhu et al. (2024) leverages its efficiency in long-sequence modeling to propose a bidirectional state space model (bi-ssm) for visual feature adaptation. However, unlike the structured grids of images, point clouds are unordered and irregular, making the learning of sequence order lead to unstable pseudo-sequential dependencies. Building upon this, Mamba3D introduces a Mamba architecture with a bi-ssm strategy to enhance the effectiveness of point cloud data processing. Inspired by this, we propose the Spiking Mamba Block (SMB), which integrates the global efficient modeling of Mamba with the energy-saving advantages of Spiking Neural Network. SMB first converts the input feature $Z$ into a binary spiking sequence $Z_1$ via the SN layer, achieving sparse synaptic accumulation, reducing feature space redundancy. SMB consists of an SSM branch and a Gate branch: for the SSM branch, the spiking sequence $Z_1$ is encoded using Linear and SN layers to obtain a new spiking sequence $Z_S$. Subsequently, features $Z_L$ and $Z_C$ are obtained via a bidirectional scanning strategy to capture bidirectional feature dependencies. The Gate branch produces a sparse spiking gating matrix $Z_G$ via Linear and SN layers, which performs feature selection through Hadamard product with $Z_L$ and $Z_C$, achieving feature selection and filtering. The product matrices are summed to obtain $Z_2$, which is finally processed by the SN layer and Linear layer to yield the SMB output $Z_3$. The entire process of SMB can be expressed as,

$$Z_1 = \text{SN}(Z), \quad Z_S = \text{SN(Lin}(Z_1)), \quad Z_G = \text{SN(Lin}(Z_1)), \tag{14,15,16}$$

$$Z_L = \text{L+SSM(Conv1d}(Z_S)), \quad Z_C = \text{C-SSM(Conv1d}(Z_S)), \tag{17,18}$$

$$Z_2 = (Z_L \otimes Z_G) + (Z_C \otimes Z_G), \quad Z_3 = \text{Lin(SN}(Z_2)), \tag{19,20}$$

here L+SSM and C-SSM denote the scanning strategies proposed by the original Mamba Gu & Dao (2023) and Mamba3D Han et al. (2024), respectively. $Lin$ represents the Linear layer, $\otimes$ represents the Hadamard product. The SMB significantly reduces feature redundancy through spiking sequence sparsification while employing a non-causal Conv1D design to eliminate temporal artifacts. By incorporating bidirectional scanning, SMB enhances model capability for complex spatial structures, along with a dynamic spiking gating mechanism that selectively activates salient features while suppressing noise/redundancy. In all, This architecture achieves a remarkable reduction in energy consumption while substantially improving both the adaptability and representational capacity for point cloud processing tasks.

| Type | Methods | Year | FLOPs | ScanObjectNN | | | ModelNet40 | |
|---|---|---|---|---|---|---|---|---|
| | | | | PB_T50_RS OA↑ | OBJ_BG OA↑ | OBJ_ONLY OA↑ | OA↑ | Energy↓ |
| ANN | PointNeXt | 22'NIPS | 3.6 | 87.7 | - | - | 94.0 | 16.6 |
| | Point-BERT | 22'CVPR | 4.8 | 83.1 | 87.4 | 88.1 | 93.2 | 22.1 |
| | Point-M2AE | 22'NIPS | 3.6 | 86.4 | 91.2 | 88.8 | 94.0 | 16.6 |
| | PTv2 | 22'NIPS | 17.1 | - | - | - | 93.7 | 78.7 |
| | PointNN | 23'CVPR | 1.0 | 87.1 | - | - | 93.8 | 4.6 |
| | PointMamba | 24'NIPS | 3.6 | 82.5 | 88.3 | 87.8 | 92.4 | 16.6 |
| | PCM | 25'AAAI | 45.0 | 88.1 | - | - | 93.4 | 207.0 |
| | SIM | 25'CVPR | 3.6 | 87.3 | 92.3 | 91.4 | 92.7 | 16.6 |
| SNN | Spike PointNet | 23'ICCV | **0.1** | 69.2 | - | - | 88.6 | **0.1** |
| | SpikePointNet | 24'NIPS | 0.4 | 64.1 | 72.2 | 76.4 | 88.2 | 0.4 |
| | P2SResLNet | 24'AAAI | 3.3 | 81.0 | 78.6 | 80.2 | 88.7 | 3.0 |
| | SPT | 25'AAAI | 14.0 | 78.0 | 82.8 | 83.4 | 91.4 | 13.3 |
| | SPM | 25'ICCV | 1.5 | 84.2 | 90.2 | 89.5 | 92.3 | 5.4 |
| | **3DSMT** *w/o vot.* | - | 1.3 | **90.4 (+6.2)** | **90.8 (+0.6)** | **89.7 (+0.2)** | **94.7 (+2.4)** | 4.3 |
| | **3DSMT** *w/ vot.* | - | 1.3 | **92.0 (+7.8)** | **92.1 (+1.9)** | **90.6 (+1.1)** | **95.2 (+2.9)** | 4.3 |

Table 1: Classification results on ModelNet40 and ScanObjectNN. '-' denotes that the model did not provide results, The units of OA, Energy and FLOPs are percentage (%), millijoule (mJ) and Gigabyte (G), respectively. '*w/o vot*' denotes the method without voting strategy, while '*w vot*' indicates testing with voting strategy applied. Among the SNN-based methods, the best results are presented in bold, and the second-best results are underlined.

## 3.4 SPIKING POSITION ENCODING

To process the positional information of point clouds in spiking neural network, we propose a learnable point cloud position encoding method. By alternately stacking Spiking Neuron (SN) layers and Multi-Layer Perceptron (MLP), this method achieves spatio-temporal feature encoding for the spatial structure of 3D point clouds. For each point $p = (x, y, z) \in \mathbb{R}^3$ in the input point cloud $P \in \mathbb{R}^{N \times 3}$, the raw coordinates are first initialized with spatio-temporal features using a SN. This layer leverages the membrane potential dynamics of neurons to capture time-dependent features of the initial spatial positions. Subsequently, an MLP performs feature transformation to enhance the representation of geometric features, outputting the final position encoding.

## 4 EXPERIMENTS

We evaluate the performance of 3DSMT classification network on ModelNet40 Wu et al. (2015) and the three main variants (PB_T50_RS, OBJ_BG, OBJ_ONLY) of ScanObjectNN Uy et al. (2019) dataset. We evaluate the 3DSMT segmentation network on the ShapeNetPart Yi et al. (2016) dataset for part segmentation. All experiments are conducted on a server equipped with an AMD EPYC 7571 2.1 GHz 32-core processor and an NVIDIA RTX 3090 GPU with 24GB memory. Our implementation is based on PyTorch and SpikingJelly Fang et al. (2023). Detailed descriptions of the datasets and experimental settings are in the *Appendix*.

### 4.1 COMPARISON WITH STATE-OF-THE-ART METHODS

We conduct a rigorous performance comparison of 3DSMT with existing ANN- and SNN-based point cloud analysis methods. For evaluation, we adopt overall accuracy (OA), Energy, FLOPS and Parameters (Para.) as metrics for classification, OA for few-shot classification, as well as category mIoU (Cat. mIoU) and Instance mIoU (Ins. mIoU) for part segmentation.

**Classification on ModelNet40.** As shown in Table 1, our 3DSMT achieves an accuracy of 94.7% when trained from scratch, and reaches a top-1 accuracy of **95.2%** after optimization by the voting strategy Liu et al. (2019)(a common post-processing technique in point cloud classification, as in Mamba3D Han et al. (2024)). Compared with the current state-of-the-art SNN-based method SPM Wu et al. (2025b), 3DSMT improves accuracy by **2.9%** while also reducing energy consumption by **1.1 mJ**. Compared with ANN-based methods, 3DSMT outperforms the single-Mamba-based method PCM Zhang et al. (2025) (93.4% OA) by **1.8%**, and reduces energy consumption by at least **48 times**. Furthermore, 3SDMT surpasses the single-Transformer-based method PTv2 Wu et al. (2022) (93.7% OA) by 1.5% and reduces energy consumption by at least 18 times. These results

| Methods | Year | Type | 5-way | | 10-way | |
|---|---|---|---|---|---|---|
| | | | 10-shot | 20-shot | 10-shot | 20-shot |
| PointNet++ | 17'NIPS | ANN | 38.5± 3.6 | 42.4± 2.9 | 23.1± 4.8 | 18.8± 5.2 |
| Transformer | 17'NIPS | ANN | 87.8± 5.2 | 93.3± 4.3 | 84.6± 5.5 | 89.4± 6.3 |
| DGCNN | 21'NIPS | ANN | 31.6± 2.8 | 40.8± 4.6 | 19.9± 2.1 | 16.9± 1.5 |
| Mamba3D | 24'MM | ANN | 92.6± 3.7 | 96.9± 2.4 | 88.1± 5.3 | 93.1± 3.6 |
| H-Emba3D | 25'arXiv | ANN | 90.5± 3.8 | 96.0± 3.2 | 86.3± 5.1 | 92.0± 3.4 |
| SpikePointNet | 24'NIPS | SNN | 52.3± 3.2 | 57.1± 3.8 | 46.2± 2.6 | 34.1± 4.2 |
| **3DSMT** | - | SNN | **92.8± 4.3** | **96.2± 3.9** | **87.2± 5.7** | **92.1± 6.1** |

Table 2: Few-shot Classification results on ModelNet40.

highlight the superior performance of the 3DSMT hybrid architecture, and demonstrate the significant potential of SNN models for point cloud analysis.

**Classification on ScanObjectNN.** We evaluate the performance of 3DSMT on three variants of the ScanObjectNN. As shown in Table 1, 3DSMT achieves the best accuracy performance with 90.4%, 90.8%, 89.7% OA on PB_T50_RS, OBJ_BG, OBJ_ONLY variants respectively, when trained from scratch on each variant. After applying the voting strategy, the OA improves to 92.0%, 92.1%, and 90.6%, surpassing the second best SNN-based method SPM Wu et al. (2025b) by **7.8%**, **1.9%**, and **1.1%**, respectively with fewer FLOPs (**1.3G**). Notably, our method demonstrates outstanding accuracy performance on the most challenging variant PB_T50_RS. Compared to ANN-based methods, 3DSMT performs comparably to SIM Bahri et al. (2025) while requiring fewer FLOPs. This not only demonstrates the effectiveness of our hybrid Transformer-Mamba design but also highlights the performance gain of SNN's unique temporal feature extraction capability.

**Few-shot Classification on ModelNet40.** We evaluate the few-shot learning ability of 3DSMT on ModelNet40. Following the few-shot learning setup of paper Sharma & Kaul (2020), we conduct experiments with $n$-way $m$-shot configurations, where $n \in \{5, 10\}$ and $m \in \{10, 20\}$. The results are reported in Table 2. When training from scratch, the 3DSMT achieves OA of 92.8%, 96.2%, 87.2% and 92.1% under the four experimental settings, respectively, outperforming the SNN-based method SpikePointNet by a large margin. Moreover, 3DSMT outperforms a series of ANN-based baseline and get the similar performance with Mamba3D Han et al. (2024). This verifies the strong transfer ability of SNN-based 3DSMT to downstream tasks under limited data conditions.

**Part Segmentation on ShapeNetPart.** We applied 3DSMT to point cloud part segmentation on the ShapeNetPart dataset. Table 3 shows that our SNN-based 3DSMT still performs well, achieving 82.7% Cat.mIoU and 85.1% Ins.mIoU—0.4% and 0.3% higher than the prior SOTA method SPM Wu et al. (2025b), respectively. The performance improvement in segmentation is smaller than in classification mainly due to dataset differences: ShapeNetPart is noise-free, fully-formed synthetic data (easy for feature extraction), while ScanObjectNN (used for classification) is real-world data with noise and occlusions, which requires more sophisticated feature

| Methods | Year | Type | Cat.mIoU | Ins.mIoU |
|---|---|---|---|---|
| PointNet++ | 17'NIPS | ANN | 81.9 | 85.1 |
| PointMLP | 22'ICLR | ANN | 84.6 | 86.1 |
| APES | 23'CVPR | ANN | 83.1 | 85.6 |
| PointMamba | 24'NIPS | ANN | 84.4 | 86.0 |
| PCM | 25'AAAI | ANN | 85.6 | 87.1 |
| SIM | 25'CVPR | ANN | 84.1 | 85.9 |
| SPT | 25'AAAI | SNN | 81.3 | 82.9 |
| SPM | 25'ICCV | SNN | 82.3 | 84.8 |
| **3DSMT** | - | SNN | **82.7(+0.4)** | **85.1(+0.3)** |

Table 3: The Part Segmentation results on ShapeNetPart.

extraction techniques. This highlights 3DSMT's strength in real-world settings. Besides, most ANN methods show little performance difference on ShapeNetPart (OA mostly 85%-87%), indicating limited distinguishability and potential drawbacks of the dataset. Even so, 3DSMT outperforms existing SNN methods and classic ANN baselines like PointNet++ Qi et al. (2017b).

**Semantic Segmentation on S3DIS.** We conduct semantic segmentation experiments on the S3DISArmeni et al. (2016) dataset, comparing our proposed SNN-based method 3DSMT with ANN-based methods and SNN-based methods in terms of mIoU, per-category IoU, and energy consumption. As shown in Table 4, 3DSMT achieves the best performance among SNN methods with a mIoU of 70.2%, outperforming E-3DSNN (67.4%) by 2.7 percentage points. meanwhile, it maintains the lowest energy consumption of 11.4 mJ, which is more energy-efficient than E-3DSNN (14.4 mJ). Among all compared methods, ANN-based PTv3 achieves the highest mIoU (73.6%) but consumes much more energy (687.7 mJ), while 3DSMT balances segmentation accuracy and energy efficiency effectively for SNN-based semantic segmentation tasks.

| Method | Type | Type | mIoU | ceil. | floor | wall | colu. | wind. | door | table | chair | sofa | book. | board | clut. | Energy |
|--------|------|------|------|-------|-------|------|-------|-------|------|-------|-------|------|-------|-------|-------|--------|
| PointNet | 17'CVPR | ANN | 41.1 | 88.8 | 97.3 | 69.8 | 3.9 | 46.3 | 10.8 | 59.0 | 52.6 | 5.9 | 40.3 | 26.4 | 33.2 | 5.5 |
| PointNet++ | 17'NIPS | ANN | 53.5 | 89.4 | 97.7 | 75.4 | 1.8 | 58.3 | 19.5 | 79.0 | 69.2 | 59.1 | 46.2 | 58.7 | 41.6 | 5.5 |
| PointCNN | 18'NIPS | ANN | 57.3 | 92.3 | 98.2 | 79.4 | 17.6 | 22.8 | 62.1 | 74.4 | 80.6 | 31.7 | 66.7 | 62.1 | 56.7 | 324.5 |
| PointNeXt | 22'NIPS | ANN | 70.5 | 94.2 | 98.5 | 84.4 | 37.7 | 59.3 | 74.0 | 83.1 | 91.6 | 77.4 | 77.2 | 78.8 | 60.6 | - |
| PCM | 25'AAAI | ANN | 63.4 | 93.3 | 96.7 | 80.6 | 35.9 | 57.7 | 60.0 | 74.0 | 87.6 | 50.1 | 69.4 | 63.5 | 55.9 | - |
| PointRWKV | 25'AAAI | ANN | 70.5 | 94.2 | 98.3 | 86.5 | 38.6 | 64.5 | 76.2 | 88.2 | 89.3 | 65.2 | 75.6 | 78.2 | 61.3 | - |
| PTv1 | 21'ICCV | ANN | 70.4 | 94.0 | 98.5 | 86.3 | 38.0 | 63.4 | 74.3 | 89.1 | 82.4 | 74.3 | 80.2 | 76.0 | 59.3 | 76.8 |
| PTv2 | 22'NIPS | ANN | 71.6 | 93.0 | 98.1 | 86.7 | 48.0 | 62.4 | 76.1 | 88.3 | 87.6 | 77.1 | 79.2 | 77.5 | 59.8 | 400.1 |
| PTv3 | 24'CVPR | ANN | 73.6 | 92.4 | 98.3 | 86.6 | 55.8 | 63.7 | 77.1 | 83.8 | 93.3 | 79.1 | 79.4 | 85.4 | 61.7 | 687.7 |
| E-3DSNN | 25'AAAI | SNN | 67.4 | 95.3 | 98.5 | 82.3 | 28.0 | 55.8 | 71.5 | 81.2 | 89.8 | 69.2 | 76.4 | 67.0 | 61.6 | 14.4 |
| 3DSMT(ours) | - | SNN | **70.2** | 88.9 | 94.2 | 82.5 | 46.8 | 62.0 | 74.4 | 85.3 | 87.3 | 77.3 | 76.9 | 77.6 | 59.8 | **11.4** |

Table 4: Semantic Segmentation Results on S3DIS Dataset. The unit of energy consumption is mJ.

**Scene Segmentation on Semantic KITTI.** We compare our SNN-based methods 3DSMT with ANN-based methods and SNN-based methods on SemanticKITTI Behley et al. (2019), with the metric being mIoU (Val/Test). As shown in the table5, 3DSMT outperforms E-3DSNN among SNN, achieving 68.1% (Val mIoU) and 71.3% (Test mIoU), which are 4.9 and 1.9 percentage points higher respectively. While ANN method PTv3 reaches the highest Test mIoU (75.5%), 3DSMT as a point cloud SNN shows competitive performance.

| Method | Type | Input | Val | Test |
|--------|------|-------|-----|------|
| SPVNAS | ANN | point | 64.7 | 66.4 |
| Cylinder3D | ANN | point | 64.3 | 67.8 |
| PTv2 | ANN | point | 70.3 | 72.6 |
| PTv3 | ANN | point | 72.3 | 75.5 |
| E-3DSNN | SNN | voxel | 63.2 | 69.4 |
| 3DSMT(our) | SNN | point | 68.1 | 71.3 |

Table 5: Scene segmentation results on Semantic KITTI Dataset.

| Methods | Training | | Inference | |
|---------|----------|--------|-----------|--------|
| | Latency | Memory | Latency | Memory |
| SPT-512 | 326ms | 9.7G | 191ms | 5.2G |
| SPT-768 | 385ms | 12.5G | 201ms | 7.3G |
| SPT-1024 | 431ms | 15.2G | 227ms | 9.5G |
| Mamba3D | 433ms | 14.9G | 256ms | 7.8G |
| 3DSMT(our) | **298ms** | **10.1G** | **142ms** | **4.6G** |

Table 6: Model Efficiency on ModelNet40 (Latency, Memory).

## 4.2 ABLATION STUDIES

To thoroughly investigate the performance for 3DSMT, we conducted a series of ablation experiments on the ModelNet40 or ScanObjectNN, examining various factors such as hybrid architecture, threshold, time steps, neighbor scale, token, ordering and bidirectional Strategies. We further conducted ablation studies on spiking neurons, point cloud size, Spiking Hybrid Block number, 3DSMT (without spiking), and efficiency in the *Appendix*. In addition, we provide details on energy consumption estimation, hardware adaptability, and further visualization results in the *Appendix*.

**Analysis on Efficiency.** We evaluate model efficiency on the ModelNet40 dataset based on two simple metrics: Latency and Memory. Table 6 shows that the Latency and Memory of ANN-based Mamba3D are both higher than those of SNN models. For the SNN-based SPT series, the two metrics rise significantly as the number of sampling points increases. Our proposed SNN-based 3DSMT has obvious efficiency advantages. It achieves the best performance among all compared methods with 298ms training Latency, 10.1G training Memory, 142ms inference Latency, and 4.6G inference Memory. This result shows that 3DSMT can effectively reduce the time and hardware resource costs of training and inference, and achieve a good balance between efficiency and performance.

**Analysis on Hybrid Architecture.** We designed a set of control experiments to test the contributions of the hybrid strategy (Transformer and Mamba modules) and SSM branch (unidirectional SSM vs. bidirectional SSM) in the hybrid architecture to accuracy and energy efficiency. **i)** As shown in Table 7, the ANN-based Full model achieves an OA of 94.9% on ModelNet40, but its energy consumption is as high as 36.3 mJ, which is significantly higher than all SNN models. Among SNN models, the No-MT model (without Transformer and Mamba) serves as the basic baseline with the lowest accuracy; although its energy consumption is only 3.3 mJ, its performance is insufficient, while the Only-T model (Transformer only) shows improved accuracy compared to the No-MT model, and the Only-MU (unidirectional SSM only) and Only-MB (bidirectional SSM only) models (Mamba only) achieve further optimized performance, proving the enhancement effect of Mamba modules, and the hybrid Full-MUT and Full-MBT models reach the highest accuracy to

| Method | Type | Hybrid Strategy | | SSM Branch | | ScanObjectNN | | | ModelNet40 | |
|---|---|---|---|---|---|---|---|---|---|---|
| | | Transformer | Mamba | Unidirection | Bidirection | OA↑(BG) | OA↑(ONLY) | OA↑(RS) | OA↑ | Energy↓ |
| Full | ANN | ✓ | ✓ | ✗ | ✓ | 91.2 | 90.4 | 91.0 | 94.9 | 36.3 |
| No-MT | SNN | ✗ | ✗ | ✗ | ✗ | 85.5 | 84.0 | 86.2 | 92.1 | **3.3** |
| Only-T | SNN | ✓ | ✗ | ✗ | ✗ | 88.0 | 86.9 | 87.7 | 93.8 | 4.1 |
| Only-MU | SNN | ✗ | ✓ | ✓ | ✗ | 89.5 | 88.2 | 88.9 | 93.8 | 4.0 |
| Only-MB | SNN | ✗ | ✓ | ✗ | ✓ | 89.6 | 88.4 | 89.2 | 94.0 | 4.0 |
| Full-MUT | SNN | ✓ | ✓ | ✓ | ✗ | 90.2 | 89.3 | 90.1 | 94.2 | 4.3 |
| Full-MBT | SNN | ✓ | ✓ | ✗ | ✓ | **90.8** | **89.7** | **90.4** | **94.7** | 4.3 |

Table 7: Analysis on hybrid spiking Mamba-Transformer architecture. BG → OBJ_BG, ONLY → OBJ_ONLY, RS → PB_T50_RS.

| $k$ | OA (%) | mAcc (%) |
|---|---|---|
| 1 | 93.6 | 90.3 |
| 2 | 93.7 | 90.7 |
| 4 | **94.7** | **91.8** |
| 6 | 94.0 | 91.1 |

| $L$ | OA (%) | mAcc (%) |
|---|---|---|
| 32 | 92.4 | 89.6 |
| 64 | 93.4 | 90.2 |
| 128 | **94.7** | **91.8** |
| 256 | 94.0 | 90.9 |

Table 8: Effect of $k$ in SLOA on ModelNet40     Table 9: Effect of $L$ tokens on ModelNet40

date—among them, Full-MBT (hybrid strategy + bidirectional SSM) performs the best among SNN models, with OA values of 89.7%, 90.4%, and 90.8% on various ScanObjectNN variants and 94.7% on ModelNet40. **ii)** The impact of SSM branch configuration is also significant: for Mamba-only models, Only-MB (bidirectional SSM) is more accurate than Only-MU (unidirectional SSM) (e.g., ModelNet40 OA rises from 93.8% to 94.0%), and for hybrid strategy models, Full-MBT (bidirectional SSM) has higher accuracy than Full-MUT (unidirectional SSM) (e.g., ModelNet40 OA increases from 94.2% to 94.7%); notably, bidirectional SSM does not add extra energy consumption, indicating that it can capture more comprehensive information and improve model performance without sacrificing energy efficiency

**Analysis on SNN Threshold and Timestep.** In spiking neural network (SNN), the threshold decides how easy it is for neurons to activate, and the timestep shows the network's ability to collect temporal information. Both are key parameters unique to SNN. We did multiple sets of experiments using different combinations of thresholds (0.5, 1.0, 1.5, 2.0) and timesteps (1, 2, 3, 4). The results are shown in Figure 2. When the threshold is fixed, increasing the timestep can make the SNN better at collecting temporal features and capturing point cloud information. However, lengthy timesteps (i.e., beyond 3) may bring in unnecessary information and degrade performance. When the timestep is the same, a too-low threshold brings in noise, and a too-high one stops useful features from working.

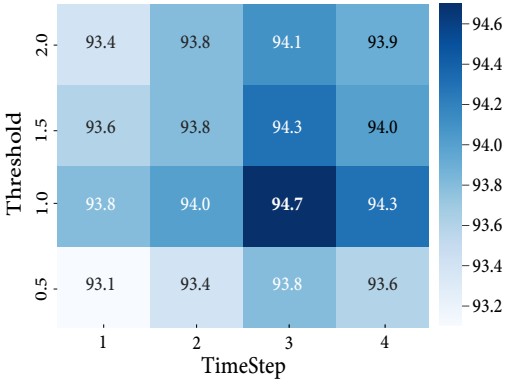

Figure 2: Analysis on threshold–timestep combinations for classification accuracy.

So the model's performance first goes up and then drops as the threshold decreases. After optimizing the experimental setting, the best parameters for the SNN are TimeStep=3 and Threshold=1.0. With this setup, the model reaches the highest overall accuracy (OA) of 94.7%.

**Effect of Neighbor Scale and Token.** We analyzed the 3DSMT model performance under different parameters. In table 8, the scale of the local neighborhood was adjusted by varying the parameter $k$ in the SLOA module. Experiments show that as $k$ increases from 1 to 4, the OA and Mean Class Accuracy (mAcc) of the model gradually improve, reaching peak values of 94.7% and 91.8% at $k = 4$. However, when $k$ exceeds 4, redundant background information is introduced, leading to performance degradation. Table 9 demonstrates that as the token sequence length $L$ increases, both OA and mAcc first rise and then fall, with optimal performance achieved at $L = 128$, beyond which accuracy reduces due to repeated feature extraction.

| Ordering Strategies | OA (%) | mAcc (%) |
|---|---|---|
| Shuffle All | 92.9 | 90.0 |
| Shuffle SSM | 93.2 | 90.4 |
| Z-order | 93.8 | 90.9 |
| No Order | **94.7** | **91.8** |

| Bidirectional Strategy | OA (%) | mAcc (%) |
|---|---|---|
| Unidirection-SSM | 94.2 | 90.9 |
| L-SSM + T-SSM | 94.3 | 91.1 |
| C-SSM + T-SSM | 94.3 | 91.4 |
| L-SSM + C-SSM | **94.7** | **91.8** |

Table 10: Effect of different ordering strategies on the ModelNet40 dataset.

Table 11: Effect of different bidirectional strategy on the ModelNet40 dataset.

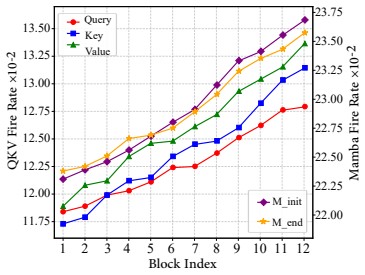

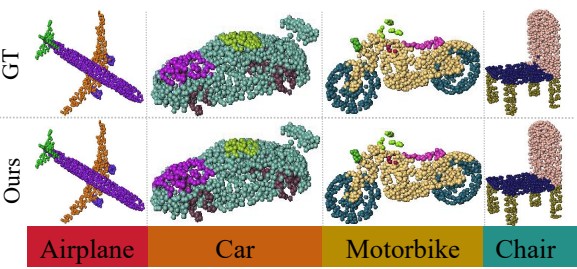

Figure 3: The firing rates of $Q$, $K$, $V$ and $M_{\text{init}}$, $M_{\text{end}}$.

Figure 4: The Part segmentation on ShapeNetPart. Top line is ground truth and bottom line is our prediction.

**Effect of Ordering and Bidirectional Strategies.** Consistent with Mamba3D's experiments, we first studied ordering strategies and found that 3DSMT still achieves optimal performance without any ordering strategy. Relevant results are in Table 10, indicating 3DSMT can effectively extract features from unordered point clouds. Based on this, we further evaluated the impact of various scanning strategies on point cloud classification accuracy in Table 11: ablation experiments on L-SSM (original Mamba), channel-flipped C-SSM (Mamba3D) and time-flipped T-SSM (SPM) showed that among unidirectional SSM strategies, Mamba3D's C-SSM (for unordered point clouds) had the highest accuracy. Moreover, bidirectional SSM maintained higher classification accuracy than unidirectional SSM, fully supporting the effectiveness of our proposed SMB module. Among bidirectional SSM strategies, the L-SSM + C-SSM combination achieved the optimal 94.7% accuracy. Hence, we adopt this bidirectional scanning strategy.

### 4.3 FURTHER ANALYSIS

**Spike Firing Rate.** Spike Firing Rate refers to the frequency at which a neuron generates spikes over a period of time, used to characterize the average activity level of the neuron (typically measured by the number of spikes per unit time), which can quantify the activation degree of the neuron. As shown in Figure 3, we recorded the firing rates of $Q$, $K$, and $V$ in the SLOA module of 3DSMT and the firing rates of the spiking layers in the initial input and final output layers of the SMB module. The extremely low firing rate leads to the sparse computation characteristics of 3DSMT.

**Visualization.** We visualize the part segmentation results of 3DSMT and the Ground Truth in Figure 4. The visualization results of 3DSMT are highly consistent with the Ground Truth, with only slight differences in fine structures such as airplane tail contours and car edges, maintaining a high overall segmentation quality. While ensuring the accuracy of 3D point cloud segmentation tasks, 3DSMT also provides a solution with significant energy efficiency advantages.

## 5 CONCLUSION

We leverage the event-driven ultra-low power consumption characteristics of spiking neural network to propose an novel hybrid architecture, 3DSMT. To address the energy consumption in processing unordered point clouds, 3DSMT hybridizes the Spiking Local Offset Attention for capturing local geometric details and the Spiking Mamba Block for linear-complexity global feature fusion, achieving an optimal balance between accuracy and efficiency. Extensive experiments show that 3DSMT outperforms existing SNN methods in point cloud shape classification, few-shot classification, and part segmentation tasks. It not only reduces energy consumption but also surpasses many traditional ANN-based models. This paper provides new insights for the development of high-performance and low-energy point cloud analysis. In the future, we will further explore the application potential of SNN in complex point cloud tasks such as object detection and scene understanding.

ACKNOWLEDGMENTS

This research was partially supported by the Australian Research Council Discovery Project (DP240101926) and the National Natural Science Foundation of China (Nos. 62206001, 52205531, U23B20105, 52575580).

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

## A    Spiking Neuron Model

Spiking neuron is the fundamental unit of SNN, we choose Integrate-and-Fire (IF) Bulsara et al. (1996) model as the spiking neuron in the proposed 3DSMT. The dynamics of a IF neuron can be formulated as follows:

$$u_t = V_{t-1} + I_t \tag{1}$$

$$S_t = \Theta(u_t - V_{th}) \tag{2}$$

$$V_t = u_t(1 - S_t) + V_{reset}S_t \tag{3}$$

Where $V_{t-1}$ represents the membrane potential of the neuron at time step $t-1$, which undergoes a spike trigger evaluation. $I_t$ represents the input current at time step $t$. $u_t$ denotes the membrane potential of the neuron after incorporating neuron dynamics at time step $t$. In Equation (2), $\Theta(\cdot)$ symbolizes the spike trigger evaluation at time step $t$. When the membrane potential $u_t$ exceeds the firing threshold $V_{th}$, the neuron emits a spike, setting $S_t = 1$; otherwise, $S_t = 0$. In Equation (3), $V_t$ represents the membrane potential of the neuron at time step $t$ after the spike evaluation. If no spike is produced, $V_t$ is equivalent to $u_t$; otherwise, it is reset to the potential $V_{reset}$.

## B    Explanation of the Permutation Invariance of the SLOA

The Spiking Local Offset Attention (SLOA) module in 3DSMT ensures permutation invariance in the local feature extraction process through three layers of strictly symmetric operation designs, making it unaffected by the sequence order of points in point clouds or patches. Specifically: First, SLOA adopts the K-Norm mechanism for local feature propagation. This process only relies on the spatial relative distance between the central point and its neighboring points, rather than the sequence order of neighboring points. Neighboring points with the same distance receive completely symmetric feature updates, eliminating order dependence at the source. Second, it aggregates neighborhood features through the K-Pool (Max-Pooling) operation. As a typical permutation-invariant operation, Max-Pooling's output result is only related to the maximum value of neighborhood features and has no correlation with the input order of features, further ensuring the stability of local aggregation results. Finally, in the offset attention calculation stage, the Query (Q), Key (K), and Value (V) matrices are all generated based on features fused with Spiking Position Encoding. Their dot-product attention calculation depends on the 3D spatial position similarity of points, and not the permutation order of Tokens in the sequence. At the same time, the offset between attention features and input features is calculated via element-wise subtraction, focusing only on feature differences rather than order relationships. These three layers of symmetric designs collectively ensure that even if the global Spiking Mamba Block (SMB) path relies on Token sequences, the local geometric features extracted by SLOA still maintain strict permutation invariance, fully adapting to the unordered characteristics of point clouds.

## C    The Rationality of Spike Mamba

We explain the rationality of Spike Mamba from two aspects: the parallel implementation of the IF neuron reset mechanism, and the time-step alignment scheme for Mamba.

### C.1    Parallel Implementation of IF Neuron Reset Mechanism

To address the compatibility issue between the IF neuron reset mechanism and parallel computing in Mamba training, this study leverages the tensor-level batch processing capability of the SpikingJelly framework, converting the reset operation into a conditional element-wise assignment logic executable in parallel on GPUs. Specifically, for the neuron membrane potential matrix $V \in \mathbb{R}^{B \times T \times N \times C}$ (where $B$ denotes batch size, $T$ denotes timestep, $N$ denotes the number of point clouds, and $C$ denotes feature dimension) at each timestep, the spike trigger matrix $S = \Theta(u - V_{th})$ is first calculated using a step function (where $u$ is the current membrane potential, and $V_{th} = 1.0$ is the optimal threshold, as shown in the ablation experiment in Section 4.2). Subsequently, parallel reset is performed based on $S$: $V_t = u_t \cdot (1 - S_t) + V_{reset} \cdot S_t$ (with $V_{reset} = 0$). This process

is implemented using the 'torch.where' function in PyTorch, which enables batch scheduling of reset operations for all neurons using the SIMD (Single Instruction, Multiple Data) architecture of GPUs, avoiding efficiency losses caused by serial loops. In the Spiking Mamba Block (SMB) of 3DSMT, this reset process is executed synchronously with the Conv1D feature encoding and bidirectional SSM (L+SSM/C-SSM) state update processes of Mamba. That is, the spike generation and reset operations are embedded into the computation chain of $Z_1 = SN(Z)$ (Equation 12) and $Z_S = SN(Lin(Z_1))$ (Equation 13), with no additional time overhead, ensuring overall training efficiency.

## C.2 ALIGNMENT SCHEME BETWEEN IF NEURON TIMESTEP AND MAMBA DISCRETE TIMESTEP

The timestep $T = 3$ of the IF neuron (used for spatio-temporal feature accumulation) and the "sequence dimension" of Mamba (corresponding to the token sequence of point cloud patches) belong to different feature processing levels. We achieve their alignment through feature dimension mapping in the Spiking Patch Embedding (SPE) module. The specific process is as follows: First, for each local patch $x_p^i \in \mathbb{R}^{k \times 3}$ (where $k$ is the number of points in the KNN neighborhood) of the input point cloud, it is duplicated into an input $E \in \mathbb{R}^{T \times k \times 3}$ with $T = 3$ timesteps. The membrane potential matrix $U \in \mathbb{R}^{T \times k \times C}$ containing spatio-temporal information is obtained through computation using the MLP and SN layers of SPE. Subsequently, the MaxPool operation (Equation 2) is applied to aggregate the features of the $T$ timesteps, resulting in a global patch feature $G \in \mathbb{R}^{k \times C}$. This operation preserves key geometric information across different timesteps while compressing the time dimension. Finally, $U$ and $G$ are concatenated and processed through MLP and SN layers to output $U' \in \mathbb{R}^{L \times C}$ (where $L$ is the number of center points sampled by FPS). This output is stacked with the learnable [CLS] token and Spiking Position Encoding ($S_{pos}$) to form a sequence of $(L+1) \times C$, which serves as the input to Mamba. Through this process, the $T = 3$ spatio-temporal features of the IF neuron are effectively converted into the token sequence dimension processable by Mamba. This not only retains the event-driven characteristics of spiking computation but also achieves full compatibility with the linear-complexity global modeling capability of Mamba, avoiding conflicts between the time dimension and the sequence dimension.

# D DATASET

ModelNet40 Wu et al. (2015) is a widely used benchmark dataset in point cloud processing, specifically designed for 3D object classification tasks. It contains 12,311 manually synthesized CAD models from 40 common furniture and daily item categories (e.g., tables, chairs, airplanes, cars, etc.). These models are generated by uniformly sampling point cloud surfaces, with each point cloud typically containing 10,000 points. ModelNet40 serves as a standard testbed for evaluating the generalization ability of point cloud classification algorithms, thanks to its clear object structures, high-quality point clouds, and moderate scale.

ScanObjectNN Uy et al. (2019) aims to bridge the gap between synthetic datasets and real-world applications by providing 3D point cloud data directly scanned from real scenes. It includes approximately 15,000 object instances across 15 categories, with three key variants offering different challenge levels: OBJ_BG (Object with Background) contains target object point clouds cropped from scenes while retaining original background points to force models to learn recognition amid noise; OBJ_ONLY (Object Only) provides 'pure' object point clouds with manually removed backgrounds, focusing on geometric identification; PB_T50_RS (Partially Bounded with Trash objects, 50% rotation, and Sensor noise)—the most challenging variant—includes point clouds with severe occlusion, background noise, irrelevant trash objects, 50% random sample rotation, and simulated sensor noise, evaluating model robustness in complex real-world messy scenarios.

ShapeNetPart Yi et al. (2016) is a subset of the ShapeNetCore Chang et al. (2015) dataset, focusing on the task of 3D object part segmentation. It is based on the ShapeNetCore dataset, with part annotations made on its models. It covers 16 object categories and contains 16,881 3D models. In addition to geometric data, the dataset also includes detailed part annotation information in the .json format. ShapeNetPart plays an important role in research on 3D object part segmentation algorithms, providing strong support for training neural networks for accurate part segmentation of objects.

S3DISArmeni et al. (2016) is a classic dataset for 3D indoor semantic segmentation, collected from three different buildings at Stanford University, covering 6 independent areas (e.g., offices, corridors, laboratories). It stores data in point cloud format, including 13 common indoor semantic categories such as floors, walls, ceilings, furniture, and electrical appliances, with high annotation accuracy and strong scene diversity. It is mainly used to evaluate the performance of 3D point cloud semantic segmentation algorithms and serves as a benchmark in indoor 3D environment understanding.

Semantic KITTI Behley et al. (2019) is an outdoor 3D semantic segmentation dataset extended from the KITTI autonomous driving dataset. Its data is collected by on-board LiDAR under various real road conditions like urban areas, rural areas, and highways. It includes continuous point cloud data of 22 sequences, defining 28 semantic categories covering core targets in autonomous driving scenarios such as vehicles, pedestrians, bicycles, roads, and buildings. Featuring strong scene dynamics, large data scale, and temporal information, it is commonly used for training and evaluating algorithms like 3D semantic segmentation and object detection in autonomous driving.

# E    MORE EXPERIMENT RESULTS

This section will provide a detailed introduction to the specific experimental setup and some supplementary experimental results.

## E.1    EXPERIMENT CONFIGURATION.

Experiments are conducted on a server equipped with an AMD EPYC 7571 2.1 GHz 32-core processor and an NVIDIA RTX 3090 GPU with 24GB video memory. Our implementation is based on PyTorch and SpikingJelly Fang et al. (2023).

**Classification on ModelNet40.** We use $N = 1024$ points as input and apply scaling and translation for data augmentation. The model is trained using the AdamW optimizer with cosine learning rate decay, an initial learning rate of 0.0005, a weight decay of 0.05, a batch size of 24, and 300 training epochs.

**Classification on ScanObjectNN.** We employ rotation as data augmentation with a point cloud size of $N = 2048$. During training, we utilize the AdamW optimizer with cosine decay, setting the initial learning rate to 0.0005, weight decay to 0.05, batch size to 24, and train for 300 epochs.

**Few-shot Classification on ModelNet40.** We adopt the ModelNet40 dataset and conduct experiments under the $n$-way $m$-shot setting, where $n$ denotes the number of randomly sampled classes and $m$ represents the number of samples per class. The model is trained using only $n \times m$ samples. During testing, 20 novel samples per class are randomly selected as test data. We evaluate combinations of $n \in \{5, 10\}$ and $m \in \{10, 20\}$, reporting the mean accuracy and standard deviation over 10 independent runs. The AdamW optimizer with cosine decay is used with an initial learning rate of 0.0005, weight decay of 0.05, batch size of 24, and 150 training epochs.

**Part Segmentation on ShapeNetPart.** We utilize input point clouds of $N = 2048$ points without normals and employ cross-entropy as the loss function. Training is performed using the AdamW optimizer with cosine decay, an initial learning rate of 0.0002, a batch size of 16, and 300 epochs. For both the S3DISArmeni et al. (2016) and Semantic KITTI Behley et al. (2019) datasets, the training configurations were set with reference to PTv3 Wu et al. (2024b) to ensure consistent and comparable experimental results.

| Methods | ModelNet40 | | ScanObjectNN(PB_T50_RS) | |
|---|---|---|---|---|
| | OA (%) | mAcc (%) | OA (%) | mAcc (%) |
| TS | **94.7** | **91.8** | 94.1 | 91.0 |
| R | 89.3 | 85.6 | **90.4** | **86.3** |

Table 12: Performance comparisons of different data augmentation methods on both the ModelNet40 and the ScanObjectNN(PB_T50_RS) datasets.

| Spiking Neurons | OA (%) | mAcc (%) |
|---|---|---|
| LIF | 93.8 | 90.9 |
| IF | **94.7** | **91.8** |

Table 13: Effect of different spiking neurons to the accuracy performance on the ModelNet40 dataset.

| Metric | Train | Test |
|---|---|---|
| Runtime | 150s | 27s |
| Memory | 10.1G | 4.6G |

Table 14: Analysis of Efficiency on the ModelNet40 dataset.

| Num | OA (%) | mAcc (%) |
|---|---|---|
| 4 | 93.7 | 90.3 |
| 8 | 93.8 | 90.6 |
| 12 | **94.7** | **91.8** |
| 16 | 93.9 | 90.7 |

Table 15: Effect of SHB Number on ModelNet40

| $N$ | OA (%) | mAcc (%) |
|---|---|---|
| 512 | 93.4 | 90.2 |
| 1024 | **94.7** | **91.8** |
| 2048 | 94.1 | 90.8 |
| 4096 | 93.8 | 90.6 |

Table 16: Effect of point size on ModelNet40

### E.2 DATA AUGMENTATION.

We define two different data augmentation methods: Translation-Scaling (TS) and Rotation (R). We also conduct comparative experiments on different datasets, with the results shown in the table 12. On ModelNet40, the 3DSMT model with TS data augmentation achieves improved performance, while the R data augmentation method performs better on the most challenging variant dataset (PB_T50_RS) of ScanObjectNN.

### E.3 EFFECT OF DIFFERENT SPIKING NEURONS.

The core of SNN is spiking neurons. The IF neuron only requires accumulation and threshold comparison, making it easy to implement on neuromorphic chips with extremely low energy consumption, especially suitable for resource-constrained edge devices. The leakage mechanism of LIF neurons requires additional circuits to simulate membrane potential decay, increasing hardware complexity and energy consumption. We conducted an ablation study on the performance impact of these two neurons on 3DSMT, and the results are shown in the Table 13. The IF neuron is more suitable for application in large-scale complex point cloud models.

### E.4 ANALYSIS ON EFFICIENCY.

We evaluate the efficiency of the 3DSMT model based on two metrics: runtime duration and memory consumption. The experimental equipment and specific configurations are described in Section Experiment Configuration. The ablation experiments adopt a setting of 3 Timestep and calculate the results for one epoch. As shown in Table 14, 3DSMT can be implemented on a single GPU with 24GB video memory, featuring relatively short training and testing times and low video memory usage, providing a new solution for large-scale point cloud processing.

### E.5 EFFECT OF SHB NUMBER.

3DSMT employs a multi-block stacked Spiking Hybrid Block (SHB) architecture. To determine the optimal number of stacked blocks, we tested configurations with 4, 8, 12, and 16 blocks. Table 15 shows that 12 stacked SHB blocks achieve the highest classification accuracy of 94.7%. We argue that increasing the number of stacked SHB blocks enhances the model's ability to capture complex relationships in point clouds and effectively utilizes the sparsity of spiking signals to improve efficiency. However, exceeding 12 blocks introduces redundant calculations, leading to a decline in generalization ability.

### E.6 EFFECT OF POINT CLOUD SIZE.

Mainstream point cloud models commonly use 1,024 uniformly sampled points. To assess the impact of sampling scale, we conducted ablation studies, revealing that 1,024 points offer the best trade-off between accuracy and efficiency (see Table 16).

| Dataset | ScanObjectNN | | |
|---|---|---|---|
| | OBJ_BG | OBJ_ONLY | PB_T50_RS |
| OA (%) | 90.8 | 89.7 | 90.4 |
| mAcc (%) | 88.3 | 87.4 | 88.0 |
| Energy (mJ) | **4.5** | **4.5** | **4.9** |

Table 17: Energy Consumption of the Three Variants of the ScanObjectNN Dataset.

### E.7 ANALYZE ON ENERGY CONSUMPTION OF THE THREE VARIANT DATASETS OF SCANOBJECTNN.

We calculated the energy consumption of the three variant datasets of ScanObjectNN (OBJ_BG, OBJ_ONLY, PB_T50_RS), with classification performance (OA and mAcc) as auxiliary evaluation metrics. The results are shown in Table 17. The energy consumption values on the three variant datasets of ScanObjectNN (OBJ_BG/OBJ_ONLY/PB_T50_RS) are 4.5 mJ, 4.5 mJ, and 4.9 mJ respectively.

### E.8 ANALYSIS ON SPIKING ARCHITECTURE.

To more comprehensively evaluate the energy efficiency of 3DSMT in point cloud analysis, we kept all structures of 3DSMT unchanged and only made one modification: removing all Spiking Neuron (SN) layers and replacing them with ordinary neurons using ReLU activation (to simulate the continuous activation characteristic of traditional Artificial Neural Network (ANN)), denoted as "3DSMT (without spiking)". The experimental results are shown in Table 18: the full-precision floating-point 3DSMT (without spiking) achieves a slightly higher accuracy than our spike-based 3DSMT. However, the accuracy loss introduced by the spiking mechanism is negligible, whereas the resulting reductions in energy consumption and computational load offer substantial advantages.

| Architecture | OA (%) | mAcc (%) | Flops (G) | Energy (mJ) |
|---|---|---|---|---|
| 3DSMT (without spiking) | **94.9** | **92.0** | 7.9 | 36.3 |
| 3DSMT (with spiking) | 94.7 | 91.8 | **1.3** | **4.3** |

Table 18: Effect of different Architecture to the accuracy performance on the ModelNet40 dataset.

## F COMPARISON TO ANN AND SNN-BASED METHODS

To clearly demonstrate the excellent balance between performance and energy consumption of 3DSMT, we conducted a multi-dimensional method comparison (as shown in Figure 5): When compared with SNN-based methods, 3DSMT achieved leading OA accuracy in multiple point cloud classification tasks such as ModelNet40(MN40), OBJ_BG (BG), OBJ_ONLY (ONLY), and PB_T50_RS (RS). Meanwhile, it met the requirements of low FLOPs and low energy consumption, breaking through the trade-off bottleneck between accuracy and efficiency of traditional SNN. When compared with ANN-based methods, 3DSMT not only surpassed most ANN methods in OA accuracy but also achieved extremely low FLOPs and energy consumption loss. The proposed 3DSMT demonstrates the comprehensive advantage of "high accuracy - low energy" in point cloud processing, providing new ideas for the design of point cloud models that balance performance and deployment requirements.

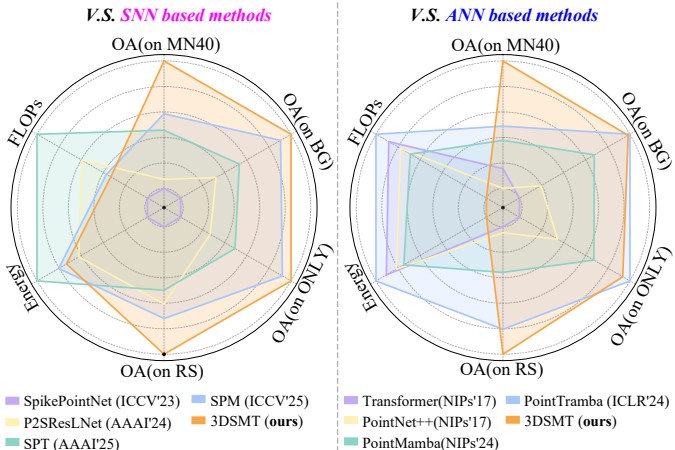

Figure 5: Comparison of our proposed method with SNN and ANN based methods in terms of energy consumption, FLOPs, overall accuracy (OA) on ModelNet40 dataset (denoted as MN40) and three variants of ScanObjectNN dataset, including OBJ_BG (denoted as BG), OBJ_ONLY (denoted as ONLY) and PB_T50_RS (denoted as RS).

## G   ENERGY CALCULATION

In artificial neural network (ANN), the number of floating-point operations (FLOPs) is an important metric for evaluating model computational complexity. In contrast, spiking neural network (SNN) use synaptic operations (SOPs) as the fundamental measure of computation. SOPs refer to the synaptic transmission process that occurs when spikes are generated. Unlike the continuous activation in ANN, SNN only trigger computations when spikes occur, significantly reducing energy consumption. The calculation formula for SOPs is as follows:

$$\text{SOPs}^l = f_r \times T \times \text{FLOPs}(l) \tag{4}$$

where $l$ denotes the index of the spiking layer in 3DSMT; $f_r$ refers spike firing rate; $T$ is the time step; $\text{FLOPs}(l)$ represents the number of floating-point operations (specifically multiply-and-accumulate operations, MAC); SOPs represents the quantity of spike-driven accumulation (AC) operations. 3DSMT is designed with biological plausibility in mind, and ideally, it can be directly deployed on neuromorphic hardware. The inference energy consumption of 3DSMT can be expressed as:

$$E_{\text{3DSMT}} = E_{\text{MAC}} \times (\text{FLOPs}^1_{\text{Conv}} + \text{FLOPs}_{\text{FC}}) + E_{\text{AC}} \times \left( \sum_{l=1}^{L} \text{SOPs}^l \right) \tag{5}$$

where $\text{FLOPs}^1_{\text{Conv}}$ represents the floating-point operations of the first floating-point convolutional layer in the Spiking Patch Embedding module, and $\text{FLOPs}_{\text{FC}}$ represents the floating-point operations of the fully connected (FC) layers in various downstream tasks. We assume that both MAC and AC operations are implemented on SpikingJelly Fang et al. (2023), where $E_{\text{MAC}} = 4.6\,\text{pJ}$ and $E_{\text{AC}} = 0.9\,\text{pJ}$ (Kundu et al., 2021b; Hu et al., 2021; Horowitz, 2014; Zhou et al., 2025; Kundu et al., 2021a).

## H   VISUALIZATION

To explore the feature representation capability of the 3DSMT model, we use t-SNE to map high-dimensional features into a 2D space for visual analysis of feature clustering effects across different datasets. As shown in Figure 6, feature points of the ModelNet40 dataset exhibit a diffused distribution with partial inter-class overlaps, though core feature clusters show obvious clustering trends. Feature points of the OBJ_BG dataset form distinct cluster separations with clear class boundaries, while those of the OBJ_ONLY dataset are highly aggregated with minimal inter-class overlap. For

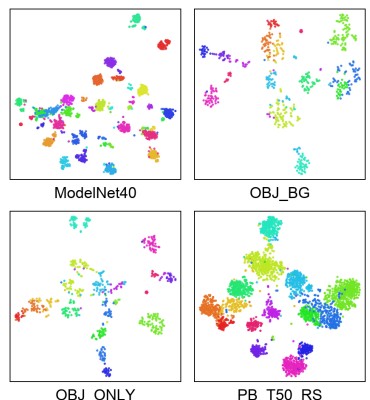

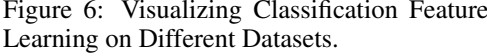

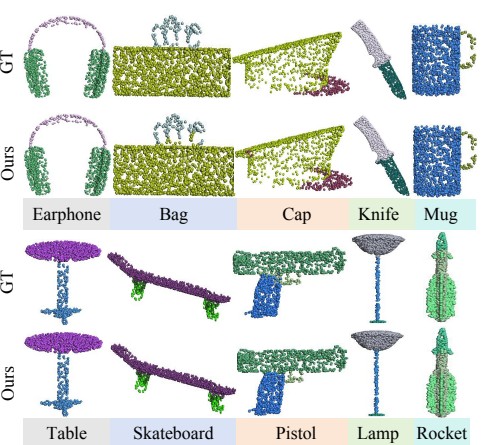

Figure 6: Visualizing Classification Feature Learning on Different Datasets.

Figure 7: Visualization of Part segmentation results on ShapeNetPart. Top line is our prediction and bottom line is ground truth.

the PB_T50_RS dataset, although feature distributions slightly diffuse due to complex noise, major class clusters maintain compact structures. These results verify the strong capability of 3DSMT in capturing features of unordered point clouds.

We also provide more part segmentation visualizations to quantitatively demonstrate the effectiveness and performance of 3DSMT, as shown in Figure 7. The part segmentation results of 3DSMT are almost identical to the ground truth across different categories. Slightly more complex categories may exhibit minor discrepancies, but these do not significantly affect the overall performance.

## I    CLAIM OF LINEAR COMPLEXITY FOR 3DSMT

The core of 3DSMT's linear complexity comes from two key architectural designs. The Spiking Local Offset Attention (SLOA) module does not use the global self-attention of standard Transformers, which has a complexity of $O(N^2)$. Instead, it only focuses on a fixed number of K-nearest neighbors for each point (K=4 in experiments). Its corresponding complexity is $O(N \times K^2)$, which simplifies to $O(N)$ because K is a constant. The Spiking Mamba Block (SMB) is based on the selective state space model (SSM) core, which inherently has a sequential processing complexity of $O(N)$. Therefore, the overall computational complexity of 3DSMT is dominated by the sum of these two linear operations, which is $O(N) + O(N) = O(N)$. This ensures the model's linear complexity with respect to the number of input points (N).

## J    NEUROMORPHIC HARDWARE RATIONALITY

We expect to directly deploy 3DSMT on neuromorphic hardware. However, designing RC circuits for processing 3D point clouds on neuromorphic hardware requires additional engineering efforts. Point cloud data typically exists in floating-point format, so it is a natural choice to adopt an Embedding method with "multiply-accumulate (MAC)" operations at the network's initial stage to generate spike signals for each point. Additionally, to pursue higher classification accuracy, the final fully connected (FC) layer needs to perform some MAC operations, which are difficult to implement on neuromorphic hardware. Although there is currently no method combining laser scanning with an event-driven mechanism to generate 3D point cloud spikes from the source, theoretically, our Spiking Local Offset Attention, Spiking Mamba Block, and Spiking Position Encoding can all be implemented on neuromorphic hardware. They can convert floating-point point clouds into spike form before performing convolution (Conv) operations and execute accumulation (AC) operations to accumulate the weights of postsynaptic neurons.

## K    LLM USAGE STATEMENT

In accordance with the policy requirements of the ICLR 2026 Conference regarding the use of Large Language Models (LLMs), the authors of this paper hereby declare that: All core academic content of this research—including the formulation of research ideas, design of methodologies, collection and analysis of experimental data, derivation of results, and refinement of conclusions—was independently completed by human authors in its entirety, with no involvement of any LLM tools.

During the paper writing process, the authors did not use LLMs for content generation, code writing, literature analysis, or result verification. Grammar checking of text expression and format adjustment were all completed manually, without assistance from any LLM tools such as ChatGPT, GPT-4, or Claude.

The authors confirm that there is no undisclosed LLM usage in this research, and all content complies with the requirements for academic integrity specified in the ICLR Code of Ethics. The authors bear full responsibility for the authenticity, accuracy, and originality of this paper, and have verified through manual checks that there are no factual errors or plagiarism risks.

## L    ETHICS STATEMENT

We confirm that all authors have read and comply with the ICLR Code of Ethics. Our work does not involve human subjects, sensitive data collection or release, potentially harmful methodologies or applications, conflicts of interest, discrimination/bias/fairness concerns, privacy/security issues, legal compliance challenges, or research integrity matters that require explicit addressing.

## M    REPRODUCIBILITY STATEMENT

To ensure the reproducibility of our results, we will make our implementation code available as supplementary material. The datasets used (e.g., ModelNet40, ScanObjectNN, ShapeNetPart for point cloud tasks) are publicly accessible, with relevant details provided in the main text. Key experimental settings (including hyperparameters like SNN threshold and timestep optimized in our analysis) and procedures are thoroughly documented in the manuscript. Any theoretical derivations supporting our work are included in the appendix.

