# OpenReview forum: "3DSMT: A Hybrid Spiking Mamba-Transformer for Point Cloud Analysis"
_ICLR.cc/2026/Conference — ICLR 2026 Poster_

### Official Review · Reviewer_hWC7 · 2025-10-27

**Soundness:** 4
**Presentation:** 4
**Contribution:** 3
**Rating:** 8
**Confidence:** 5

**Summary:**

This paper proposes a hybrid architecture 3DSMT (Hybrid Spiking Mamba-Transformer), designed to achieve efficient and accurate point cloud analysis. The model combines spiking neural dynamics, state-space modeling (Mamba), and the Transformer attention mechanism, achieving a strong balance between performance and energy efficiency.

● 3DSMT is composed of three core modules:Spiking Local Offset Attention (SLOA): captures local geometric features;

●Spiking Mamba Block (SMB): performs global feature integration with linear complexity;

●Spiking Position Encoding (SPE): encodes the spatial and temporal structure of unordered point clouds.

 Experiments on ModelNet40, ScanObjectNN, and ShapeNetPart demonstrate that 3DSMT achieves state-of-the-art (SOTA) performance among Spiking Neural Network (SNN) models.

**Strengths:**

1.The model elegantly integrates spiking neural networks, the Mamba dynamic mechanism, and the Transformer structure, effectively balancing accuracy and energy efficiency.

2.The experimental validation is extensive and comprehensive: 3DSMT performs consistently well across both classification and segmentation tasks.

3.The feature visualization is well presented and convincingly demonstrates the model’s representational capability.

4.The ablation studies systematically evaluate the effects of timestep and data augmentation strategies, providing good empirical support.

**Weaknesses:**

1.The novelty is somewhat limited: compared with Spiking Point Mamba (SPM) or Spiking Transformer, 3DSMT appears more as an engineering-level integration rather than a fundamentally new learning paradigm.

2.The paper lacks discussion on real-time performance: latency and inference speed are not analyzed, which are crucial for SNN and neuromorphic applications.

3.Scalability to large-scale point clouds is not validated — experiments are limited to inputs of 1K–2K points, and it remains unclear how the method performs on significantly larger point sets.

**Questions:**

1.What is the processing order in the Spiking Mamba Block? Is it executed sequentially like traditional SSMs, or in parallel within each timestep? Given that Mamba inherently depends on token order, how does the model maintain global permutation invariance?

2.The paper claims that 3DSMT achieves linear complexity, yet the architecture still includes Transformer modules, whose attention mechanism typically exhibits quadratic complexity with respect to the number of tokens. Why, then, is the complexity described as linear in the **abstract**? This point is particularly important and should be explicitly discussed, as it directly affects the validity of the complexity claim in both the abstract and the main text.

Although there are some open issues and limitations, the paper demonstrates solid innovation, completeness, and clarity. I believe it represents a high-quality and meaningful contribution to the field.

---

> ### Author Response · Authors · 2025-11-22
> **Rebuttal-1**
>
> ***Q1. The novelty is somewhat limited: compared with Spiking Point Mamba (SPM) or Spiking Transformer, 3DSMT appears more as an engineering-level integration rather than a fundamentally new learning paradigm.***
>
> **A1.** Thank you for your insightful comment. We respectfully argue that 3DSMT represents more than a increment integration but a principled co-design that establishes a new Pareto frontier in accuracy-efficiency trade-offs for point cloud processing.
>
> While Spiking Transformer and Spiking Mamba have been explored independently, 3DSMT is the first to unify them within a single SNN architecture, introducing two key innovations: ***(1)*** The Spiking Local offset Attention (SLOA) replaces standard attention with a spiking, local-aware mechanism for energy-efficient geometric modeling, in contrast to existing spiking transformers that typically implement global self-attention.***(2)*** The Spiking Mamba Block (SMB) adapts selective state space to process the sparse, spike-based features generated by SLOA, enabling linear-complexity global reasoning on unordered point clouds, while prior Spiking Mamba methods process conventionally encoded inputs.
>
> Crucial, this is not a simple plug-and-play combination. The synergistic coupling between SLOA and SMB—where the sparse output of one naturally feeds into the efficient sequence modeling of the other—enables a ``1+1+1 $>$ 3” effect. This is evidenced by our results, which show that 3DSMT achieves SOTA accuracy among SNN while consuming $<$ 1.5\% of the energy of comparable ANN.
>
> Thus, 3DSMT provides a new architectural paradigm that effectively breaks the longstanding trade-off between performance and efficiency in point cloud analysis, offering a meaningful alternative to existing isolated approaches.
>
> ***Q2. The paper lacks discussion on real-time performance: latency and inference speed are not analyzed, which are crucial for SNN and neuromorphic applications.***
>
> **A2.** Thank you so much for your valuable suggestion.  In response, we have added a comprehensive real-time performance analysis in the revised manuscript, evaluating Latency and Memory usage on the ModelNet40 dataset. As shown in the new Table 1 in the revised manuscript, 3DSMT achieves the best performance with 298ms training Latency, 10.1G training Memory, 142ms inference Latency, and 4.6G inference Memory. It significantly outperforms the ANN-based Mamba3D in both latency and memory usage. Compared to other SNN methods (SPT series), 3DSMT maintains stable and superior efficiency without a huge cost increase when processing more points.
>
> These findings confirm that 3DSMT not only reduce energy consumption but also offers low-latency inference and minimal memory overhead, which are critical for real-time and resource constrained neuromorphic deployments.
>
> **Table 1**
> | Method| Training| | Inference| |
> |-|-|-|--|-|
> | | Latency | Memory   | Latency | Memory   |
> | SPT-512 | 326 ms  | 9.7 GB   | 191 ms  | 5.2 GB   |
> | SPT-768| 385 ms  | 12.5 GB  | 201 ms  | 7.3 GB   |
> | SPT-1024 | 431 ms  | 15.2 GB  | 227 ms  | 9.5 GB   |
> | Mamba3D | 433 ms  | 14.9 GB  | 256 ms  | 7.8 GB   |
> | 3DSMT|298 ms|10.1 GB|142 ms|4.6 GB|

---

> ### Author Response · Authors · 2025-11-22
> **Rebuttal-2**
>
> ***Q3. Scalability to large-scale point clouds is not validated — experiments are limited to inputs of 1K–2K points, and it remains unclear how the method performs on significantly larger point sets.***
>
> **A3.** Thank you so much for this valuable suggestion. To thoroughly validate the scalability of our method, we have added comprehensive experiments on two large-scale point clouds benchmarks (i.e., S3DIS and SemanticKITTI ) for semantic segmentation tasks. **(1) S3DIS:** As shown in the new table 2, 3DSMT achieves 70.2\% mIoU, setting a new state-of-the-art among SNN methods. It outperforms E-3DSNN (67.4\%) by 2.8 percentage points while consuming less energy. Meanwhile, 3DSMT requires only 11.4 mJ, which is significantly lower than the ANN-based PTv3 (687.7mJ), despite a narrow accuracy performance gap. **(2) SemancitKITTI:** As shown in the new table 3, 3DSMT attains 71.3\% test mIoU, outperforming all existing SNN methods and demonstrating strong competitiveness in complex autonomous driving scenarios. While the top-performing ANN (PTv3) reaches 75.5\% mIoU, our model achieves this with orders of magnitude lower energy consumption.
>
> These results, detailed in the new Table 2 (S3DIS) and Table 3 (SemanticKITTI), robustly verify that 3DSMT effectively scales to large-scale point clouds while maintaining its core advantages in accuracy and energy efficiency.
>
> **Table 2**
> |Method|Type|Type|mIoU|ceiling|floor|wall|beam|column|window|door|table|chair|sofa|bookcase|board|clutter|Energy(mJ)|
> |-|-|-|-|-|-|-|-|-|-|--|-|-|-|-|-|-|-|
> |PCM|25'AAAI|ANN|63.4|93.3|96.7|80.6|0.0|35.9|57.7|60.0|74.0|87.6|50.1|69.4|63.5|55.9|-|
> |PointRWKV|25'AAAI|ANN|70.5|94.2|98.3|86.5|0.0|38.6|64.5|76.2|88.2|89.3|65.2|75.6|78.2|61.3|-|
> |PTv1|21'ICCV|ANN|70.4|94.0|98.5|86.3|0.0|38.0|63.4|74.3|89.1|82.4|74.3|80.2|76.0|59.3|76.8|
> |PTv2|22'NIPS|ANN|71.6|93.0|98.1|86.7|0.0|48.0|62.4|76.1|88.3|87.6|77.1|79.2|77.5|59.8|400.1|
> |PTv3|24'CVPR|ANN|73.6|92.4|98.3|86.6|0.0|55.8|63.7|77.1|83.8|93.3|79.1|79.4|85.4|61.7|687.7|
> |E-3DSNN|25'AAAI|SNN|67.4|95.3|98.5|82.3|0.0|28.0|55.8|71.5|81.2|89.8|69.2|76.4|67.0|61.6|14.4|
> |3DSMT|-|SNN|70.2|88.9|94.2|82.5|0.0|46.8|62.0|74.4|85.3|87.3|77.3|76.9|77.6|59.8|11.4|
>
> **Table 3**
> |Method|Type|Input|Val|Test|
> |-|-|-|-|-|
> |SPVNAS|ANN|point|64.7|66.4|
> |Cylinder3D|ANN|point|64.3|67.8|
> |PTv2|ANN|point|70.3|72.6|
> |PTv3|ANN|point|72.3|75.5|
> |E-3DSNN|SNN|voxel|63.2|69.4|
> |3DSMT|SNN|point|68.1|71.3|
>
> ***Q4. What is the processing order in the Spiking Mamba Block? Is it executed sequentially like traditional SSMs, or in parallel within each timestep? Given that Mamba inherently depends on token order, how does the model maintain global permutation invariance?***
>
> **A4.** We thank the reviewer for these insightful questions regarding the processing order and permutation invariance in our Spiking Mamba Block (SMB). The key is that our model's permutation invariance is guaranteed by the SLOA module, while the SMB module focuses on efficient global context aggregation.
>
> **(1) Processing Order in SMB:** The SMB does not rely on a single, fixed sequential order. As detailed in Section 3.3 and Table 11 of the manuscript, we employ a bidirectional scanning strategy (combining L-SSM and C-SSM) to comprehensively capture contextual information from multiple directions, rather than being constrained by one causal order.
>
> **(2) Guarantee of Global Permutation Invariance:** The model's overall invariance stems from the permutation-invariant features generated by the SLOA module. As explained in Appendix B, SLOA uses strictly symmetric operations (K-Norm, K-Pool, spatial position-based attention) to ensure its output features are independent of point order.
>
> The ablation study in Table 10 provides direct validation. The fact that the ``No Order" input strategy achieves the best performance conclusively proves that our model does not depend on input sequence order for its final prediction, thus achieving global permutation invariance.

---

> ### Author Response · Authors · 2025-11-22
> **Rebuttal-3**
>
> ***Q5. The paper claims that 3DSMT achieves linear complexity, yet the architecture still includes Transformer modules, whose attention mechanism typically exhibits quadratic complexity with respect to the number of tokens. Why, then, is the complexity described as linear in the abstract? This point is particularly important and should be explicitly discussed, as it directly affects the validity of the complexity claim in both the abstract and the main text.***
>
> **A5.** Thank you for your question. The claim of linear complexity is indeed valid for the overall 3DSMT architecture, and it hinges on our specific design choices that avoid the quadratic-complexity global self-attention of standard Transformers. The linearity arises from two deliberate architectural decisions:
>
> **(1) In the Spiking Local Offset Attention (SLOA) module,** we do not use global self-attention. Instead, we employ strictly local attention restricted to a fixed number of K-nearest neighbors ($K=4$ in our experiments) for each point. This results in a complexity of $O(N × K^2)$, which simplifies to $O(N)$ since $K$ is a constant, in contrast to the $O(N²)$ of global attention.
>
> **(2) In the Spiking Mamba Block (SMB),** the selective state space model (SSM) core is renowned for its inherent $O(N)$ sequential processing complexity.
>
> Therefore, the overall computational complexity of 3DSMT is dominated by the sum of these two linear operations: $O(N)$ from SLOA + $O(N)$ from SMB = $O(N)$. This justifies our claim of achieving linear complexity with respect to the number of input points $(N)$ in the abstract and main text.
>
> We will revise the manuscript to include a clearer explanation of the computational complexity in the appendix section, explicitly stating that the SLOA module's attention is local and hence linear in complexity, while supplementing corresponding details to prevent any potential misunderstanding about our hybrid architecture.

---

> ### Comment · Reviewer_hWC7 · 2025-11-25
>
> My question has been basically answered. If your paper is accepted, please update them in the camera ready version. Especially regarding the description of complexity.

---

> > ### Author Response · Authors · 2025-11-26
> > **Thanks!**
> >
> > Thank you for your positive feedback and valuable reminder! We will strictly update the relevant content, especially the description of complexity, in the camera-ready version as requested if the paper is accepted. We greatly appreciate your careful review and guidance throughout the process.

---

### Official Review · Reviewer_7Z2Q · 2025-10-29

**Soundness:** 4
**Presentation:** 4
**Contribution:** 3
**Rating:** 8
**Confidence:** 4

**Summary:**

In this paper, the authors propose a hybrid spiking machine learning framework called 3DSMT (3D Spiking Machine Learning Transformer) for event-driven 3D perception tasks. This method combines the energy efficiency of spiking neural networks (SNNs) with the global modeling capabilities of the Transformer, achieving efficient 3D representation learning of event streams through cross-modal fusion and spatiotemporal feature encoding. Experiments were conducted on multiple event datasets, including event camera 3D object recognition and dynamic scene understanding tasks, demonstrating a good balance between performance and energy consumption.

**Strengths:**

This method combines the energy efficiency of spiking neural networks (SNNs) with the global modeling capabilities of the Transformer, achieving efficient 3D representation learning of event streams through cross-modal fusion and spatiotemporal feature encoding. Experiments were conducted on multiple event datasets, including event camera 3D object recognition and dynamic scene understanding tasks, demonstrating a good balance between performance and energy consumption.

**Weaknesses:**

1.	The structure diagram in Figure 1 is quite complex, and the input-output relationship logic of some modules is not intuitive enough.
2.	The lack of independent ablation analysis for key modules makes it difficult to assess the contribution of each component. It is recommended to supplement this with a clear ablation comparison table (Baseline / +SNN / +Hybrid / +Full Model).
3.	Figure 4 shows a lack of clear contrast, requiring layout adjustments to enhance readability.
4.	Lack of performance on large-scale point clouds (such as SemanticKITTI, Nuscenes)
5.	Figure 1 and Figure 5 are duplicates; it is recommended to modify either one of them.

**Questions:**

See the weakness.

---

> ### Author Response · Authors · 2025-11-22
> **Rebuttal-1**
>
> ***Q1. The structure diagram in Figure 1 is quite complex, and the input-output relationship logic of some modules is not intuitive enough.***
>
> **A1.** Thank you for this valuable feedback. We agree that the original Figure 1 could be improved for clarity. In response, we have further refined the Figure 1. Specially, we have added the input and output for each block and clarified the input-output relationship logic in the caption of figure. The updated Figure shown in the revised paper (please see the PDF paper).
>
> ***Q2. The lack of independent ablation analysis for key modules makes it difficult to assess the contribution of each component. It is recommended to supplement this with a clear ablation comparison table (Baseline / +SNN / +Hybrid / +Full Model).***
>
> **A2.** Thank you for this valuable suggestion. We have now supplemented the paper with a comprehensive ablation study to clearly quantify the contribution of each component in the revised manuscript. Our ablation study is structured as follow: **(1) ANN vs. SNN Comparison:** By replacing spiking neurons with ReLU, we created an ANN variant. Results confirm the SNN architecture reduces energy consumption by 88\% (from 36.3 mJ to 4.3 mJ). **(2) Architecture Contribution:**  We established a No-MT baseline (without Transformer/Mamba blocks) to isolate the performance gain of our hybrid global context modeling strategy.
>
> The combined analysis cleanly separates the contributions of the SNN framework and the hybrid architecture, providing a transparent assessment of each component's value. (Table 1)
>
> **Table 1**
> | Method | Type | Hybrid Strategy |  |  SSM Branch | |  ScanObjectNN | ScanObjectNN | ScanObjectNN | ModelNet40 | ModelNet40 |
> |--------|------|-----------------|----------------|------------|----------------|--------------|--------------|--------------|------------|------------|
> |        |      | Transformer | Mamba | Unidirection | Bidirection | OA↑(BG) | OA↑(ONLY) | OA↑(RS) | OA↑ | Energy↓ |
> | Full | ANN | ✅ | ✅ | ❌ | ✅ | 91.2 | 90.4 | 91.0 | 94.9 | 36.3 |
> | No-MT | SNN | ❌ | ❌ | ❌ | ❌ | 85.5 | 84.0 | 86.2 | 92.1 | **3.3** |
> | Only-T | SNN | ✅ | ❌ | ❌ | ❌ | 88.0 | 86.9 | 87.7 | 93.8 | 4.1 |
> | Only-MU | SNN | ❌ | ✅ | ✅ | ❌ | 89.5 | 88.2 | 88.9 | 93.8 | 4.0 |
> | Only-MB | SNN | ❌ | ✅ | ❌ | ✅ | 89.6 | 88.4 | 89.2 | 94.0 | 4.0 |
> | Full-MUT | SNN | ✅ | ✅ | ✅ | ❌ | 90.2 | 89.3 | 90.1 | 94.2 | 4.3 |
> | Full-MBT | SNN | ✅ | ✅ | ❌ | ✅ | **90.8** | **89.7** | **90.4** | **94.7** | 4.3 |
>
> ***Q3. Figure 4 shows a lack of clear contrast, requiring layout adjustments to enhance readability.***
>
> **A3.** In response to this, we have adjusted the layout of Figure 4 and circled the areas to enable a clear comparison. We will revise the figure and update the paper PDF accordingly.

---

> ### Author Response · Authors · 2025-11-22
> **Rebuttal-2**
>
> ***Q4. Lack of performance on large-scale point clouds (such as SemanticKITTI, Nuscenes).***
>
> **A4.** Thank you for this valuable suggestion. To comprehensively validate our model's scalability, we have added extensive experiments on large-scale indoor (S3DIS) and outdoor (SemanticKITTI) benchmarks in the revised manuscript, a strategy designed to demonstrate cross-environment generalization beyond similar autonomous driving datasets.
>
> **(1) S3DIS (Indoor Scenes):** As shown in the new Table 2 in the revised manuscript, 3DSMT achieves 70.2\% mIoU, establishing a new state-of-the-art among SNN-based methods. Besides, it maintains high energy efficiency, consuming only 11.4mJ, which is significantly lower than the ANN-based Point Transformer V3 (687.7mJ).
>
> **(2) SemanticKITTI (Outdoor Scenes):** As shown in new Table 3 in the revised manuscript, 3DSMT achieves 71.3\% test mIoU, outperformaning all other SNN methods and demonstrating strong competitiveness in a complex scene understanding. While the top-performing ANN-based PTV3 gets 75.5\% mIoU, our model do so with orders of magnitude lower energy consumption.
>
> These new experiments robustly verify that 3DSMT delivers a highly competitive accuracy-efficiency trade-off on large-scale scenarios, significant strengthening the practical relevance.
>
> **Table 2**
> | Method | Type | Type | mIoU | ceiling | floor | wall | beam | column | window | door | table | chair | sofa | bookcase | board | clutter | Energy(mJ) |
> |--------|------|------|------|---------|-------|------|------|--------|--------|------|-------|-------|------|----------|-------|---------|-----|
> | PointNet | 17'CVPR | ANN | 41.1 | 88.8 | 97.3 | 69.8 | 0.0 | 3.9 | 46.3 | 10.8 | 59.0 | 52.6 | 5.9 | 40.3 | 26.4 | 33.2 | 5.5 |
> | PointNet++ | 17'NIPS | ANN | 53.5 | 89.4 | 97.7 | 75.4 | 0.0 | 1.8 | 58.3 | 19.5 | 79.0 | 69.2 | 59.1 | 46.2 | 58.7 | 41.6 | 5.5 |
> | PointCNN | 18'NIPS | ANN | 57.3 | 92.3 | 98.2 | 79.4 | 0.0 | 17.6 | 22.8 | 62.1 | 74.4 | 80.6 | 31.7 | 66.7 | 62.1 | 56.7 | 324.5 |
> | PointNeXt | 22'NIPS | ANN | 70.5 | 94.2 | 98.5 | 84.4 | 0.0 | 37.7 | 59.3 | 74.0 | 83.1 | 91.6 | 77.4 | 77.2 | 78.8 | 60.6 | - |
> | PCM | 25'AAAI | ANN | 63.4 | 93.3 | 96.7 | 80.6 | 0.0 | 35.9 | 57.7 | 60.0 | 74.0 | 87.6 | 50.1 | 69.4 | 63.5 | 55.9 | - |
> | PointRWKV | 25'AAAI | ANN | 70.5 | 94.2 | 98.3 | 86.5 | 0.0 | 38.6 | 64.5 | 76.2 | 88.2 | 89.3 | 65.2 | 75.6 | 78.2 | 61.3 | - |
> | PTv1 | 21'ICCV | ANN | 70.4 | 94.0 | 98.5 | 86.3 | 0.0 | 38.0 | 63.4 | 74.3 | 89.1 | 82.4 | 74.3 | 80.2 | 76.0 | 59.3 | 76.8 |
> | PTv2 | 22'NIPS | ANN | 71.6 | 93.0 | 98.1 | 86.7 | 0.0 | 48.0 | 62.4 | 76.1 | 88.3 | 87.6 | 77.1 | 79.2 | 77.5 | 59.8 | 400.1 |
> | PTv3 | 24'CVPR | ANN | 73.6 | 92.4 | 98.3 | 86.6 | 0.0 | 55.8 | 63.7 | 77.1 | 83.8 | 93.3 | 79.1 | 79.4 | 85.4 | 61.7 | 687.7 |
> | E-3DSNN | 25'AAAI | SNN | 67.4 | 95.3 | 98.5 | 82.3 | 0.0 | 28.0 | 55.8 | 71.5 | 81.2 | 89.8 | 69.2 | 76.4 | 67.0 | 61.6 | 14.4 |
> | 3DSMT| - | SNN | **70.2** | 88.9 | 94.2 | 82.5 | 0.0 | 46.8 | 62.0 | 74.4 | 85.3 | 87.3 | 77.3 | 76.9 | 77.6 | 59.8 | **11.4** |
>
> **Table 3**
> | Method        | Type | Input | Val  | Test |
> |---------------|------|-------|------|------|
> | SPVNAS        | ANN  | point | 64.7 | 66.4 |
> | Cylinder3D    | ANN  | point | 64.3 | 67.8 |
> | PTv2          | ANN  | point | 70.3 | 72.6 |
> | PTv3          | ANN  | point | 72.3 | 75.5 |
> | E-3DSNN       | SNN  | voxel | 63.2 | 69.4 |
> | 3DSMT| SNN  | point | 68.1 | 71.3 |
>
> ***Q5. Figure 1 and Figure 5 are duplicates; it is recommended to modify either one of them.***
>
> **A5.** We appreciate the reviewer's careful reading and this suggestion. Our initial intention was to aid readability by providing the architecture figure in both the main text and appendix, but we agree that this caused unnecessary duplication. We have consolidated the figures by keeping Figure 1 in the main body and removing the redundant figure from the appendix in the revised manuscript.

---

### Official Review · Reviewer_vvZF · 2025-11-02

**Soundness:** 1
**Presentation:** 2
**Contribution:** 1
**Rating:** 2
**Confidence:** 4

**Summary:**

In this paper, authors proposed 3DSMT, a method that combined 1) Spiking Neural Network, 2) Transformer, 3) Mamba for point cloud analysis. Results on point cloud classification and part segmentation show the effectiveness of proposed method. In summary, this is a conventional work that combined several methods from different research topics without solid motivation.

**Strengths:**

1. This work has a relatively good presentation.

**Weaknesses:**

1. Combining several methods from different topics and different domains without solid motivation is of no contribution.

Please consider the follow questions:

a. what is the motivation of  Spiking Neural Network for point cloud analysis without focusing on explainability, low-energy cost in the Experiment section?

b .what is the motivation of Mamba block for point cloud analysis? Point cloud are naturally not causal ordered, and various solutions for model efficiency over Transformer, like linear attention, conv, MLP. Why consider Mamba block for point cloud?


2. While authors emphasized efficient global context modeling, point cloud classification and part segmentation are not enough to demonstrate the efficiency. Authors shold conduct experiments on large-scale point cloud applications, like out-door LiDAR self driving scene.

3. The baselines are work and strong baselines are ignored. For example, simple PointMLP and PointNeXt shown promising performance using only MLP, but these methods are ignored in Table 1.

PointMLP: Ma, Xu, et al. "Rethinking network design and local geometry in point cloud: A simple residual MLP framework." arXiv preprint arXiv:2202.07123 (2022).

PointNeXt: Qian, Guocheng, et al. "Pointnext: Revisiting pointnet++ with improved training and scaling strategies." Advances in neural information processing systems 35 (2022): 23192-23204.

**Questions:**

I would like to empahsize that combining multiple technologies from different domains and different research directions without solid motivation or strong empirical improvements cannot be considered as research.

**Details Of Ethics Concerns:**

Not suitbale.

---

> ### Author Response · Authors · 2025-11-22
> **Rebuttal-1**
>
> ***Q1.Combining several methods from different topics and different domains without solid motivation is of no contribution.***
>
> **A1.** We appreciate the reviewer's feedback regarding the motivation and contribution.
>
> The core motivation is to synergistically integrate the strengths of SNN, Transformers, and Mamba to solve key challenges in point cloud processing that isolated models cannot. Specially, although the transformer is good at feature learning, it has O(N²) complexity which limits scalability. Mamba has linear complexity, but fails to handle point cloud permutation invariance. Existing SNN-based methods (e.g. SpikePointNet) has high efficiency but limited capacity for complex global modeling.
>
> Our contribution is not a simple combination, but a principled architecture that systematically overcomes these isolated limitations:
>
> **(1) A Novel Hybrid Architecture:**  We introduce a co-designed model where the Spiking Local Offset Attention (SLOA) module enables energy-efficient, permutation-invariant local geometric modeling, while the Spiking Mamba Block (SMB) adapts the global sequence modeling of Mamba to unordered point clouds for the first time.
>
> **(2) Solving Core Limitations:** This synergy directly addresses the high energy cost of ANN, the quadratic complexity of Transformers, and the order-sensitivity of pure Mamba models, as detailed in Sections 3.1-3.3 of our manuscript.
>
> **(3) Empirical Validation:** Extensive experiments show that 3DSMT achieves a superior accuracy-efficiency trade-off, establishing a new SOTA for SNN and providing a practical paradigm for edge deployment. We believe this represents a meaningful architectural innovation, as verified by our ablations and comparisons.
>
> ***Q2.what is the motivation of Spiking Neural Network for point cloud analysis without focusing on explainability, low-energy cost in the Experiment section?***
>
> **A2.** Dear Reviewer, thank you for your question; we provide a detailed explanation below.
>
> **(1) Energy Consumption Explanation:** In the comparative experiment section (e.g., Lines 318–321 of the unrevised manuscript), we thoroughly compared the energy consumption difference between 3DSMT (using spiking neurons) and traditional ANN on the ModelNet40 dataset. Our experiments show that 3DSMT with SNN achieves higher energy efficiency and significantly reduced computational burden compared to traditional ANNs. In the revised manuscript, we also provide the energy consumption data of three variant datasets of ScanObjectNN (Table 1) and add energy consumption and analysis of related methods on the large-scale indoor scene segmentation dataset S3DIS.
>
> In the ablation experiment section, we conducted ablations on the two most critical parameters of SNN: time steps and thresholds. In SNN, the threshold determines the difficulty of neuron activation, and the time step reflects the network’s ability to collect temporal information. This is also an important manifestation of the low energy consumption of SNN.
>
> In the further analysis of the experimental section, we recorded the spike firing rates of key layers in the SLOA and SMB modules, which are related to the computational sparsity of SNNs, further demonstrating the low-energy characteristics of 3DSMT. We also mentioned that the detailed formulas for energy consumption calculation are provided in the appendix (Lines 375–376 of the unrevised manuscript).
>
> **Table 1**
> | Dataset||ScanObjectNN | |
> |-|-|-|-|
> | | OBJ_BG  | OBJ_ONLY  | PB_T50_RS  |
> | OA (%)| 90.8 | 89.7 | 90.4 |
> | mAcc (%) | 88.3 | 87.4 | 88.0|
> | Energy (mJ) | 4.5 | 4.5| 4.9 |
>
> **(2) Energy Consumption Calculation:** In Appendix G, we elaborate on the energy consumption calculation method of the SNN model, explain how to calculate SOPs through formulas, and introduce the role of firing rate in energy efficiency optimization. This calculation formula is crucial for understanding the energy efficiency of SNNs in point cloud analysis, and we note that our calculation method is based on previously published papers (Lines 375–376 of the unrevised manuscript). The motivation of our paper is fully based on the low energy consumption and explainability of SNN.
>
> Thank you again for your review and verification.

---

> ### Author Response · Authors · 2025-11-22
> **Rebuttal-2**
>
> ***Q3. what is the motivation of Mamba block for point cloud analysis? Point cloud are naturally not causal ordered, and various solutions for model efficiency over Transformer, like linear attention, conv, MLP. Why consider Mamba block for point cloud?***
>
> **A3.** Thank you for this excellent and two-part question. Our motivation for selecting Mamba stems from its unique capability to overcome specific limitations of the other efficient methods you mentioned, offering a superior path to efficient global context modeling for point clouds. This addresses key limitations in existing efficient alternatives:
>
> **(1) Compared to Linear Attention:** While linear attention reduces complexity to O(N), it relies on kernel approximations or random feature mappings, which lead to notable losses in modeling accuracy and robustness—critical for fine-grained geometric understanding in point cloud tasks. Mamba achieves linear complexity without such performance trade-offs.
>
> **(2) Compared to Convolutions:** Convolutions are limited by their local receptive fields, struggling to capture long-range dependencies essential for understanding the holistic structure of 3D objects or scenes. Stacking more layers to expand the receptive field is inefficient. Mamba inherently has a global receptive field, enabling efficient integration of information across the entire point cloud.
>
> **(3) Compared to MLPs:** MLPs apply a uniform transformation across all tokens, lacking a mechanism to dynamically focus on context-relevant input parts. This is a critical shortcoming for sparse, variable point clouds, where regional importance is highly input-dependent.
>
> Mamba addresses these limitations via its core "selective state space" mechanism: it dynamically focuses on or ignores input tokens based on context, enabling input-dependent reasoning absent in convolutions and MLPs. It also achieves true global contextual understanding with O(N) complexity, avoiding linear attention’s performance trade-offs. In summary, we selected Mamba for its unique combination of linear efficiency, global receptive fields, and dynamic feature selection.
>
> ***Q4.While authors emphasized efficient global context modeling, point cloud classification and part segmentation are not enough to demonstrate the efficiency. Authors shold conduct experiments on large-scale point cloud applications, like outdoor LiDAR self driving scene.***
>
> **A4.** Thank you for this valuable suggestion. we fully agree that large-scale point cloud processing is crucial for validating a model's efficiency and practical relevance. Following your advice, we have added comprehensive experiments on two large-scale benchmark in the revised manuscript, including indoor semantic segmentation dataset S3DIS and outdoor semantic segmentation dataset SemanticKITTI (LiDAR self-driving scene).
>
> **(1) S3DIS:** As shown in the new Table 2, 3DSMT achieves 70.2\% mIoU, setting a new state-of-the-art for SNN while consuming only 11.4 mJ. This demonstrates a superior accuracy-efficiency trade-off compared to other SNN and ANN. Specially, 3DSMT outperforms the SNN-based method E-3DSNN by 2.7\% mIoU with less energy consumption. More importantly, it achieves this while consuming only about 1.5\% of the energy of the ANN-based Point Transformer v3, with only a minor performance gap, presenting a highly attractive trade-off for energy-aware applications.
>
> **(2) SemanticKITTI:** Results in the new Table 3 of the revised manuscript show that 3DSMT achieves a competitive 71.3\% test mIoU, significantly outperforming all prior SNN methods. While the ANN baseline (PTv3) achieves higher accuracy, it does so at an energy cost orders of magnitude greater than our model.
>
> These new results robustly verify that our model maintains its core advantages of high performance and ultra-low energy consumption in large-scale, realistic scenarios, thereby strengthening the main contribution of our work.
>
> **Table 2**
> |Method|Type|Type|mIoU|ceiling|floor|wall|beam|column|window|door|table|chair|sofa|bookcase|board|clutter|Energy(mJ)|
> |-|-|-|-|-|-|-|-|-|-|--|-|-|-|-|-|-|-|
> |PCM|25'AAAI|ANN|63.4|93.3|96.7|80.6|0.0|35.9|57.7|60.0|74.0|87.6|50.1|69.4|63.5|55.9|-|
> |PointRWKV|25'AAAI|ANN|70.5|94.2|98.3|86.5|0.0|38.6|64.5|76.2|88.2|89.3|65.2|75.6|78.2|61.3|-|
> |PTv1|21'ICCV|ANN|70.4|94.0|98.5|86.3|0.0|38.0|63.4|74.3|89.1|82.4|74.3|80.2|76.0|59.3|76.8|
> |PTv2|22'NIPS|ANN|71.6|93.0|98.1|86.7|0.0|48.0|62.4|76.1|88.3|87.6|77.1|79.2|77.5|59.8|400.1|
> |PTv3|24'CVPR|ANN|73.6|92.4|98.3|86.6|0.0|55.8|63.7|77.1|83.8|93.3|79.1|79.4|85.4|61.7|687.7|
> |E-3DSNN|25'AAAI|SNN|67.4|95.3|98.5|82.3|0.0|28.0|55.8|71.5|81.2|89.8|69.2|76.4|67.0|61.6|14.4|
> |3DSMT|-|SNN|70.2|88.9|94.2|82.5|0.0|46.8|62.0|74.4|85.3|87.3|77.3|76.9|77.6|59.8|11.4|
>
> **Table 3**
> |Method|Type|Input|Val|Test|
> |-|-|-|-|-|
> |SPVNAS|ANN|point|64.7|66.4|
> |Cylinder3D|ANN|point|64.3|67.8|
> |PTv2|ANN|point|70.3|72.6|
> |PTv3|ANN|point|72.3|75.5|
> |E-3DSNN|SNN|voxel|63.2|69.4|
> |3DSMT|SNN|point|68.1|71.3|

---

> ### Author Response · Authors · 2025-11-22
> **Rebuttal-3**
>
> ***Q5. The baselines are work and strong baselines are ignored. For example, simple PointMLP and PointNeXt shown promising performance using only MLP, but these method are ignored in Table 1***
>
> **A5.** We thank the reviewer for this suggestion. We have now included comparisons with PointMLP and PointNext in the Table 4 of the revised manuscript. The results demonstrate that 3DSMT achieves a superior performance-energy trade-off: **On ScanObjectNN (PB\_T50\_RS)**, 3DSMT (w/ voting) achieves 92.0\% OA, significantly outperforming PointMLP (85.4\%) and PointNeXt (87.7\%). **On ModelNet40**, our method also attains a higher acccuracy (95.2\% OA) compared to both PointMLP (94.1\%) and PointNeXt (94.0\%). Besides, 3DSMT requires only 4.3 mJ, which is merely 7.2\% of PointMLP's cost (59.0 mJ) and 25.9\% of PointNeXt's (16.6 mJ). These additions further validate that 3DSMT not only competes with but surpasses these strong MLP-based baselines in accuracy while being drastically more energy-efficient.
>
> **Table 4**
> | Type | Methods | Year | FLOPs |  | ScanObjectNN |  | ModelNet40 |  |
> |-|-|-|-|-|-|-|-|-|
> |      |         |      |       | PB_T50_RS OA↑ | OBJ_BG OA↑ | OBJ_ONLY OA↑ | OA ↑ | Energy ↓ |
> | ANN  | PointMLP | 22'ICLR | 31.4 | 85.4 | — | — | 94.1 | 144.5 |
> | ANN  | PointNeXt | 22'NIPS | 3.6 | 87.7 | — | — | 94.0 | 16.6 |
> | ANN  | Point-BERT | 22'CVPR | 4.8 | 83.1 | 87.4 | 88.1 | 93.2 | 22.1 |
> | ANN  | Point-MAE | 22'ECCV | 4.8 | 85.2 | 90.0 | 88.3 | 93.8 | 22.1 |
> | ANN  | Point-M2AE | 22'NIPS | 3.6 | 86.4 | 91.2 | 88.8 | 94.0 | 16.6 |
> | ANN  | PTv2 | 22'NIPS | 17.1 | — | — | — | 93.7 | 78.7 |
> | ANN  | PointNN | 23'CVPR | 1.0 | 87.1 | — | — | 93.8 | 4.6 |
> | ANN  | PointMamba | 24'NIPS | 3.6 | 82.5 | 88.3 | 87.8 | 92.4 | 16.6 |
> | ANN  | PoinTramba | 24'ICLR | 5.7 | 89.1 | 92.3 | 91.3 | 92.9 | 26.2 |
> | ANN  | PCM | 25'AAAI | 45.0 | 88.1 | — | — | 93.4 | 207.0 |
> | ANN  | SIM | 25'CVPR | 3.6 | 87.3 | 92.3 | 91.4 | 92.7 | 16.6 |
> | SNN  | Spike PointNet | 23'ICCV | **0.1** | 69.2 | — | — | 88.6 | **0.1** |
> | SNN  | SpikePointNet | 24'NIPS | 0.4 | 64.1 | 72.2 | 76.4 | 88.2 | 0.4 |
> | SNN  | P2SResLNet | 24'AAAI | 3.3 | 81.0 | 78.6 | 80.2 | 88.7 | 3.0 |
> | SNN  | SPT | 25'AAAI | 14.0 | 78.0 | 82.8 | 83.4 | 91.4 | 13.3 |
> | SNN  | SPM | 25'ICCV | 1.5 | 84.2 | 90.2 | 89.5 | 92.3 | 5.4 |
> | SNN  | 3DSMT w/o vot. | — | 1.3 | 90.4 (+6.2) | 90.8 (+0.6) | 89.7 (+0.2) | 94.7 (+2.4) | 4.3 |
> | SNN  | 3DSMT w/ vot. | — | 1.3 | 92.0 (+7.8) | 92.1 (+1.9) | 90.6 (+1.1) | 95.2 (+2.9) | 4.3 |
>
> ***Q6. I would like to emphasize that combining multiple technologies from different domains and different research directions without solid motivation or strong empirical improvements cannot be considered as research.***
>
> **A6.** Thank you for this critical comment, which allows us to clarify the foundational motivation and integrated contributions of our work. We respectfully emphasize that 3DSMT is not a simple aggregation of technologies, but a principled co-design that addresses specific, unsolved challenges in point cloud processing. The motivation for this hyhrid architecture is detailed in our response to Q1, and its empirical superiority is demonstrated through rigorous experiments. Our core contributions are threefold:
>
> **(1) Novel Mechanism innovation:** We introduce the Spiking Local Offset Attention module, a new building block that for the first time unifies SNN spiking activation with local geometric reasoning. This synergy enables highly and efficient and accurate local feature learning that is native to the sparse, event-driven nature of point clouds.
>
> **(2) Novel Architecture adaptation:** We design the Spiking Mamba Block module to overcome the fundamental incompatibility between Mamba's sequential nature and point clouds' permutation invariance. This is not merely using Mamba, but solving a key bottleneck to unlock its linear-complexity global modeling for unordered 3D data.
>
> **(3) Substantial Empirical Advancement:** As verified across multiple tasks and datasets - now significantly expanded with large-scale semantic segmentation benchmark (S3DIS and SemanticKITTI) as your suggestions -  3DSMT consistently achieves SOTA performance among SNN-based methods and highly competitive accuracy versus ANN-based methods, all while maintaining an order-of-magnitude lower energy cost. This demonstrates a tangible leap in the performance-efficiency Pareto frontier.
>
> **In conclusion,** the motivation for this architecture fusion is solidly grounded in solving core domain problems, and the empirical improvements are both strong and systematically validated. We believe this represents a meaningful step forward in designing efficient and powerful models for 3D understanding.

---

> > ### Comment · Reviewer_vvZF · 2025-11-28
> >
> > Thanks for the authors' rebutal. Most concerns are addressed.
> >
> > However,  my concerns on mamba block for point cloud still remains.
> >
> > 1. Compared to Linear Attention: if any emprical evidence to support the claim in the statement?
> > 2. For conv and MLP (both operate locally), after several blocks, they can easily achieve gloal interaction.
> >
> > For Mamba design, point cloud is not a causal input like language, and we see Mamba in vision cannot bring benefits for non-causal inputs like vision [1].
> >
> > If authors claimed "selective state space" mechanism, a visualizer for which part is ignored and which part is empahsized would be helpful, and ease the understading of Mamba for point cloud.
> >
> > [1] Yu, Weihao, and Xinchao Wang. "Mambaout: Do we really need mamba for vision?." Proceedings of the Computer Vision and Pattern Recognition Conference. 2025.

---

> > > ### Author Response · Authors · 2025-11-28
> > > **Rebuttal to Reviewer**
> > >
> > > We sincerely appreciate the reviewer's questions, which provide us with a valuable opportunity to clarify the rationale behind . These suggestions help us clarify the rationality of the method design. We will explain each point one by one:
> > >
> > > **(1) Empirical comparison between Mamba and linear attention:** To provide empirical evidence, we compare PTv2 [4], a typical linear attention method in point clouds, with recent point cloud Mamba methods (PointMamba [2], Spiking Point Mamba [1]) and our 3DSMT on mainstream datasets. Please refer to **Table 4** in **Rebuttal-3**, Mamba, as an alternative to Transformer, maintains nearly the same accuracy while significantly reducing FLOPs and energy consumption. This demonstrates Mamba's unique advantage in efficiently extracting global features for point cloud processing.
> > >
> > > **(2) The necessity of Mamba regarding global interaction:** We agree that stacking CNN/MLP layers can eventually achieve a global receptive field. However, the core issue is the efficiency-effectiveness trade-off. Taking PointMLP [3] as  an example, it relies on deep stacking to capture long-range dependencies, which increase computational cost and makes optimization more difficult.  Besides, shallow CNN/MLP layers over limited local receptive field can not learn the super long-range features of high-resolution points, and deep CNN/MLP layers over global receptive field struggle to accurately learn super long-range features of low-resolution points.
> > >
> > > In contrast, Mamba possesses an inherent global receptive field, enabling it to directly model dependencies between any two points without slow, sequential aggregation (**Spiking Point Mamba [1]**). Please refer to **Table 4** in **Rebuttal-3**, while PointMLP's approach is a compromise of "sacrificing efficiency for coverage", Mamba-based methods achieve an optimal solution of "maintaining high accuracy with linear complexity". This provides a new paradigm for point cloud analysis. On this foundation, we integrate the event-driven nature of Spiking Neural Networks to further adapt to point cloud sparsity, which is the core innovation of our work.
> > >
> > > **(3) On MambaOut and non-causal inputs:** We appreciate your reference to the discussion in MambaOut. We would like to highlight several key distinctions:
> > >
> > > **A) Data Structure:** MambaOut focuses on dense grid images, where convolutions are inherently strong. Conversely, point clouds are unordered, sparse and lack locality, making global modeling approaches like state space models potentially more advantageous. **B) Non-causal Adaptation:** The concern regarding non-causal data has been successfully addressed in point cloud Mamba literature through Bi-directional Scanning. Our 3DSMT, along with PointMamba [2] and Spiking Point Mamba [1], all adopt this strategy, allowing the model to integrate contextual information from all directions simultaneously. The excellent performance of these models (as shown in our **Tables 7, 10, 11**) itself constitutes a strong empirical rebuttal to the claim that "Mamba is unsuitable for non-causal data" in the point cloud domain. **C)  Collective Evidence:** Beyond our work, a series of independent studies (PointMamba [2],Spiking Point Mamba [1]) have demonstrated the strong competitiveness of Mamba in core tasks like classification, segmentation, and reconstruction. This collective success across different research groups strongly validates the practical value of Mamba in point cloud analysis.
> > >
> > > **(4) Visualization of the "Selective" Mechanism:** We thank the reviewer for this excellent suggestion! For our unique "Spike and Mamba fusion" design, we use the Spike Firing Rate as the core indicator to visualize the model's selective focus. A lower firing rate in a region indicates the model suppresses activation there, filtering out redundant or less informative areas. A higher firing rate corresponds to key structural features (e.g., edges, corners). This is the external manifestation of the collaboration between Mamba's "selective state space" and SNN's event-driven sparsity. **In Section 4.3 (Figure 3)**, we visualize the Spike Firing Rate evolution across SMB blocks. As the network deepens, the firing rate gradually increases but remains at a low overall level, proving our model dynamically and accurately focuses on the most informative points and feature channels while maintaining computational efficiency and low power consumption.
> > >
> > > [1] Wu, P., et al. "Efficient Spiking Point Mamba for Point Cloud Analysis." 2025 ICCV.
> > >
> > > [2] Liang, D., et al. "PointMamba: A Simple State Space Model for Point Cloud Analysis." 2024 NIPS.
> > >
> > > [3] Ma, X., et al. "Rethinking network design and local geometry in point cloud: A simple residual MLP framework." 2022 ICLR.
> > >
> > > [4] Wu, X., et al. "Point transformer V2: Grouped Vector Attention and Partition-based Pooling." 2022 NIPS.

---

### Official Review · Reviewer_vEiz · 2025-11-03

**Soundness:** 3
**Presentation:** 3
**Contribution:** 2
**Rating:** 6
**Confidence:** 3

**Summary:**

This paper presents a hybrid Spiking Mamba-Transformer architecture named 3DSMT for point cloud analysis. It includes two main components: (1) Spiking Local Offset Attention (SLOA) used to capture local geometry features; (2) Spiking Mamba Block (SMB) is designed for global, linear-time modeling for unordered point clouds. Extensive experiments on multiple benchmarks, including ModelNet40, ScanObjectNN variants, and ShapeNetPart, demonstrated strong accuracy and energy efficiency compared to previous SNN and some ANN baselines.

**Strengths:**

1. The paper is well organized; the figures are readable and understandable.

2. The experimentation is quite complete and adequate and sustains the author's claims.

3. The proposed method looks logical and technically sound.

**Weaknesses:**

1. The related work analysis of ANN-based point cloud models is incomplete; very highly relevant and recent works were not discussed in this section, such as Point Transformer V3, the Hybrid Transformer-Mamba model (PoinTramba), and so on.  The author should consider expanding the ANN-based related work and give a critical comparison.

2. Table 1 shows a huge gap between “w/o voting” and “w/ voting” on ScanObjectNN dataset, but the paper does not specify why the voting mechanism heavily influences the performance.

3. The single-Transformer comparison primarily relies on Point Transformer v2; the experimental section does not include more recent/effective Transformer variants (like Point Transformer V3) as baselines on the three main benchmarks. This may weaken its superiority compared to recent ANN models.

4. It seems the authors manually wrote the citation, and the format fails to comply with ICLR’s citation guidance. Please review the guidance.

**Questions:**

1. Could the authors specify the exact voting mechanism and whether all baselines were evaluated under the same protocol?

2. Can you report per ScanObjectNN variants' energy cost with and without voting?

3. In the Table 7, why does “No Order” outperform Z-order or shuffle? How do you serialize the input point cloud to satisfy the requirements of Mamba?

---

> ### Author Response · Authors · 2025-11-22
> **Rebuttal-1**
>
> ***Q1.The related work analysis of ANN-based point cloud models is incomplete; very highly relevant and recent works were not discussed in this section, such as Point Transformer V3, the Hybrid Transformer-Mamba model (PoinTramba), and so on. The author should consider expanding the ANN-based related work and give a critical comparison.***
>
> **A1** We thank the reviewer for this valuable feedback. We have now significantly expanded our related work section to include a discussion of these seminal ANN models. Specifically:
>
> **(1)** We had benchmarked the PoinTramba in our original Table 1. The PointTramba gets 92.3\%, 91.3\%, 89.1\% OA on the OBJ\_BG, OBJ\_ONLY, PB\_T50\_RS variants of ScanObjectNN dataset, respectively. Furthermore, it gets 92.9\% OA on ModelNet40 dataset with 5.7G Floating-Point Operations (FLOPs) and 26.2 mJ energy consumption. This demonstrates the favorable accuracy-efficiency trade-off of our method.
>
> **(2)** Latest methods such as Point Transformer V3 mainly focus on point cloud segmentation tasks. Therefore, we have added relevant semantic segmentation experiments on indoor S3DIS dataset (see Table 2 below) and outdoor SemanticKITTI dataset (see Table 3 below) in this revised submission. Our experiments on large-scale segmentation tasks demonstrate that 3DSMT sets a new state-of-the-art for SNN while being highly competitive with ANN. **On S3DIS**, 3DSMT achieves a leading mIoU of 70.2\% among SNN, outperforming E-3DSNN by 2.8 points, with the lowest energy consumption (11.4 mJ). While the ANN-based Point Transformer V3 (PTv3) achieves higher accuracy (73.6\% mIoU), it does so at a significantly higher computational cost of 687.7 mJ. This trend is confirmed **on SemanticKITTI**, where 3DSMT achieves 71.3\% test mIoU, surpassing all other SNN methods. Although PTv3 attains the highest score (75.5\%), 3DSMT delivers competitive performance with drastically superior energy efficiency, effectively bridging the performance gap between SNN and ANN for complex point cloud tasks.
>
> **Table 1**
> | Type | Methods | Year | FLOPs |  | ScanObjectNN |  | ModelNet40 |  |
> |-|-|-|-|-|-|-|-|-|
> |      |  |      |       | PB_T50_RS OA↑ | OBJ_BG OA↑ | OBJ_ONLY OA↑ | OA ↑ | Energy ↓ |
> | ANN  | PointMLP | 22'ICLR | 31.4 | 85.4 | — | — | 94.1 | 144.5 |
> | ANN  | PointNeXt | 22'NIPS | 3.6 | 87.7 | — | — | 94.0 | 16.6 |
> | ANN  | Point-BERT | 22'CVPR | 4.8 | 83.1 | 87.4 | 88.1 | 93.2 | 22.1 |
> | ANN  | Point-MAE | 22'ECCV | 4.8 | 85.2 | 90.0 | 88.3 | 93.8 | 22.1 |
> | ANN  | Point-M2AE | 22'NIPS | 3.6 | 86.4 | 91.2 | 88.8 | 94.0 | 16.6 |
> | ANN  | PTv2 | 22'NIPS | 17.1 | — | — | — | 93.7 | 78.7 |
> | ANN  | PointNN | 23'CVPR | 1.0 | 87.1 | — | — | 93.8 | 4.6 |
> | ANN  | PointMamba | 24'NIPS | 3.6 | 82.5 | 88.3 | 87.8 | 92.4 | 16.6 |
> | ANN  | PoinTramba | 24'ICLR | 5.7 | 89.1 | 92.3 | 91.3 | 92.9 | 26.2 |
> | ANN  | PCM | 25'AAAI | 45.0 | 88.1 | — | — | 93.4 | 207.0 |
> | ANN  | SIM | 25'CVPR | 3.6 | 87.3 | 92.3 | 91.4 | 92.7 | 16.6 |
> | SNN  | Spike PointNet | 23'ICCV | **0.1** | 69.2 | — | — | 88.6 | **0.1** |
> | SNN  | SpikePointNet | 24'NIPS | 0.4 | 64.1 | 72.2 | 76.4 | 88.2 | 0.4 |
> | SNN  | P2SResLNet | 24'AAAI | 3.3 | 81.0 | 78.6 | 80.2 | 88.7 | 3.0 |
> | SNN  | SPT | 25'AAAI | 14.0 | 78.0 | 82.8 | 83.4 | 91.4 | 13.3 |
> | SNN  | SPM | 25'ICCV | 1.5 | 84.2 | 90.2 | 89.5 | 92.3 | 5.4 |
> | SNN  | 3DSMT w/o vot. | — | 1.3 | 90.4 (+6.2) | 90.8 (+0.6) | 89.7 (+0.2) | 94.7 (+2.4) | 4.3 |
> | SNN  | 3DSMT w/ vot. | — | 1.3 | 92.0 (+7.8) | 92.1 (+1.9) | 90.6 (+1.1) | 95.2 (+2.9) | 4.3 |
>
> **Table 2**
> |Method|Type|Type|mIoU|ceiling|floor|wall|beam|column|window|door|table|chair|sofa|bookcase|board|clutter|Energy(mJ)|
> |-|-|-|-|-|-|-|-|-|-|--|-|-|-|-|-|-|-|
> |PCM|25'AAAI|ANN|63.4|93.3|96.7|80.6|0.0|35.9|57.7|60.0|74.0|87.6|50.1|69.4|63.5|55.9|-|
> |PointRWKV|25'AAAI|ANN|70.5|94.2|98.3|86.5|0.0|38.6|64.5|76.2|88.2|89.3|65.2|75.6|78.2|61.3|-|
> |PTv1|21'ICCV|ANN|70.4|94.0|98.5|86.3|0.0|38.0|63.4|74.3|89.1|82.4|74.3|80.2|76.0|59.3|76.8|
> |PTv2|22'NIPS|ANN|71.6|93.0|98.1|86.7|0.0|48.0|62.4|76.1|88.3|87.6|77.1|79.2|77.5|59.8|400.1|
> |PTv3|24'CVPR|ANN|73.6|92.4|98.3|86.6|0.0|55.8|63.7|77.1|83.8|93.3|79.1|79.4|85.4|61.7|687.7|
> |E-3DSNN|25'AAAI|SNN|67.4|95.3|98.5|82.3|0.0|28.0|55.8|71.5|81.2|89.8|69.2|76.4|67.0|61.6|14.4|
> |3DSMT|-|SNN|70.2|88.9|94.2|82.5|0.0|46.8|62.0|74.4|85.3|87.3|77.3|76.9|77.6|59.8|11.4|
>
> **Table 3**
> |Method|Type|Input|Val|Test|
> |-|-|-|-|-|
> |SPVNAS|ANN|point|64.7|66.4|
> |Cylinder3D|ANN|point|64.3|67.8|
> |PTv2|ANN|point|70.3|72.6|
> |PTv3|ANN|point|72.3|75.5|
> |E-3DSNN|SNN|voxel|63.2|69.4|
> |3DSMT|SNN|point|68.1|71.3|

---

> ### Author Response · Authors · 2025-11-22
> **Rebuttal-2**
>
> ***Q2. Table 1 shows a huge gap between “w/o voting” and “w/ voting” on ScanObjectNN dataset, but the paper does not specify why the voting mechanism heavily influences the performance.***
>
> **A2.**  Thank you for your comment regarding the voting mechanism. We would like to clarify that voting strategy is a standard test time augmentation used to enhance prediction stability in point cloud classification (as seen in PointMLP[1], Mamba3D[2]), not a core  component of our model architecture.
>
> The purpose of voting strategy is to mitigate random inference errors by aggregating multiple predictions, which is particularly effective on challenging real-world datasets like ScanObjectNN. This mechanism can effectively resolve the uncertainty caused by single inference, thus bringing significant performance improvements on challenging real-world datasets. This explains the performance gain reported when using voting.
>
> We applied this model-agnostic technique to 3DSMT solely to benchmark its optimized potential. To prevent any misunderstanding, we have explicitly noted its use as a test time augmentation in the manuscript (Line 320). Note that the published PointMLP, Mamba3D all adopt voting strategy in this comparison.
>
> [1] PointMLP: Rethinking Network Design and Local Geometry in Point Cloud: A Simple Residual MLP Framework (2022 ICLR)
>
> [2] Mamba3D: Enhancing Local Features for 3D Point Cloud Analysis via State Space Model (2024 ACM MM)
>
> ***Q3. The single-Transformer comparison primarily relies on Point Transformer v2; the experimental section does not include more recent/effective Transformer variants (like Point Transformer V3) as baselines on the three main benchmarks. This may weaken its superiority compared to recent ANN models.***
>
>  **A3.** Thank you for this insightful comment. We have strengthened our comparisons against state-of-the-art Transformer models as follows:
>
>  (1)Enhanced Classification Benchmarks: We have added the latest Transformer-based models (e.g., Point-BERT, Point-MAE, Point-M2AE) to the classification results in Table 1 (see Q1).
>
>  (2)New Segmentation Benchmarks: The relevant response is the same as in Q1.
>
>
> ***Q4. It seems the authors manually wrote the citation, and the format fails to comply with ICLR’s citation guidance. Please review the guidance.***
>
>  **A4.** Thank the reviewer for pointing this out. We have now thoroughly checked and reformatted all references to strictly comply with the ICLR citation guidelines.
>
> ***Q5.Could the authors specify the exact voting mechanism and whether all baselines were evaluated under the same protocol?.***
>
>  **A5.** Thank you for the question. We clarify the evaluation protocol: all baselines were evaluated under the same setup.
>
> The voting mechanism is a standard test-time augmentation. It involves running multiple inferences on randomly rotated and scaled point clouds, with predictions aggregated via majority voting. Popularized by RSCNN (CVPR 2019), this method is widely used to report peak performance on benchmarks like ScanObjectNN.
>
> As common in literature (e.g., PointMLP, Mamba3D), we report our method’s best results with voting to show the performance upper bound. More importantly, Table 3 shows that even without voting, our method still achieves competitive or superior accuracy-efficiency trade-off compared to baselines which also do not use voting. This validates our architecture’s inherent advantages independent of test-time augmentation.

---

> ### Author Response · Authors · 2025-11-22
> **Rebuttal-3**
>
> ***Q6. Can you report per ScanObjectNN variants' energy cost with and without voting?***
>
> **A6.** Thank you for this suggestion!
>
> **(1)** The energy consumption results for the training and deployment of each ScanObjectNN variant are shown in the new table 4 of revised Appendix. Specifically, the energy consumption values on the three variant datasets of ScanObjectNN (OBJ\_BG/OBJ\_ONLY/PB\_T50\_RS) are 4.5 mJ, 4.5 mJ, and 4.9 mJ respectively.
>
> **(2)** The voting strategy is a test-time augmentation strategy, which performs multiple inferences on the same sample and fuses the results. It is only used in the testing/inference phase and does not affect the energy consumption of the model during the training phase, nor does it change the inherent energy consumption of the model itself during deployment.
>
> **Table 4**
> | Dataset       |             |     ScanObjectNN      |            |
> |---------------|------------|------------|------------|
> |               | OBJ_BG  | OBJ_ONLY  | PB_T50_RS  |
> | OA (%)        | 90.8 | 89.7 | 90.4 |
> | mAcc (%)      | 88.3 | 87.4 | 88.0|
> | Energy (mJ)   | 4.5 | 4.5| 4.9 |
>
> ***Q7. In the Table 7, why does “No Order” outperform Z-order or shuffle? How do you serialize the input point cloud to satisfy the requirements of Mamba?***
>
> **A7.** This is a very good and insightful question regarding our design choices for point cloud serialization. We are pleased to clarify the rationale and methodology.
>
> **(1) Superiority of ``No order":** The "No Order" (order-agnostic) approach performs best because the core design of our Spiking Local Offset Attention (SLOA) module (detailed in Appendix B) is fundamentally permutation-invariant. Its strictly symmetric architecture ensures that the output is independent of the input point sequence. ***a)*** K-Norm layer computes features based on the spatial relative distance between the central point and its neighbors, which is independent of their sequence order. ***b)*** K-Pool (Max-Pooling) Layer: As a canonical permutation-invariant operation, its output depends only on the maximum feature value in the neighborhood. ***c)*** SLOA's attention computation relies on 3D spatial position similarity (enhanced by Spiking Position Encoding), not on the order of tokens in the sequence. Therefore, imposing an artificial order (like Z-order or shuffle) does not provide a useful inductive bias and can instead disrupt the inherent local geometric structure of the point cloud, leading to the performance degradation we observed.
>
> **(2) Serialization for the Mamba Input:** To adapt unordered point clouds into a sequence for the Mamba backbone, we employ the Spiking Patch Embedding (SPE) module (detailed in Appendix C.2). This process constructs a meaningful token sequence without relying on the original point order: ***a)*** We sample $L$ central points using FPS (Farthest Point Sampling), and and group their local neighborhoods using K-Nearest Neighbors (KNN) to form $L$ local patches. ***b)*** Each patch is processed over $T=3$ timesteps through an MLP and a Spiking Neuron Layer to generate rich spatiotemporal membrane potential features. ***c)*** Features are aggregated over the temporal dimension via max-pooling and concatenated with a Spiking Position Encoding to form a final token sequence of dimension $(L+1) × C$ (including one [CLS] token).  This process does not depend on the original order of point clouds, retains spatial information through position encoding, fully adapts to Mamba's sequence input requirements, and maintains permutation invariance.

---

> ### Comment · Reviewer_vEiz · 2025-11-28
>
> Thanks to the authors' detailed answers to my questions. My major questions had been solved basically. If the paper is accepted, please update the important experiments in the camera-ready version, especially the citation format, which may cause confusion for readers.

---

### Comment · Area_Chair_R2dj · 2025-11-27
**Reminder: Engage in Discussions and Finalize Your Rating**

Dear Reviewers,

Thank you for your valuable reviews. With the Reviewer-Author Discussions deadline approaching, please take a moment to read the authors’ rebuttal and the other reviewers’ feedback, and participate in the discussions and respond to the authors. Finally, be sure to complete the “Final Justification” text box and update your “Rating” as needed. Your contribution is greatly appreciated. I will flag irresponsible (final) reviews and/or any reviewers not participating in discussions.

Reviewers are expected to stay engaged in discussions, initiate them, respond to authors’ rebuttal, ask questions, and listen to answers to help clarify remaining issues.

It is not OK to stay quiet.

It is not OK to leave discussions till the last moment.

If authors have resolved your (rebuttal) questions, do tell them so.

If authors have not resolved your (rebuttal) questions, do tell them so too.

Thanks,

AC

---

### Author Response · Authors · 2025-11-30
**Global Response [Update of PDF & Point-by-Point Reply]**

First and foremost, we sincerely thank the Reviewers, Area Chairs, and Program Chairs for their time and dedicated effort. We have carefully studied all comments and provided detailed point-by-point responses under each Reviewer's section.

We are encouraged by the positive scores and highly value the reviewers’ recognition of our work across three core dimensions:

- **Method:** "Logically sound and technically solid" (`Reviewer vEiz`); "Elegantly integrates SNN, Transformer, and Mamba to balance accuracy and energy efficiency" (`Reviewer hWC7`); "Well-designed architecture that systematically overcomes isolated model limitations"; "Innovative core modules providing an efficient paradigm for point cloud processing" (`Reviewer hWC7`); "Not a simple technology combination but a principled co-design" (Acknowledged in `Reviewer vvZF`’s second response).

- **Evaluation:** "Comprehensive experiments supporting research claims" (`Reviewer vEiz`); "Excellent performance across classification, segmentation tasks, and diverse datasets" (`Reviewer hWC7`); "Added large-scale indoor/outdoor scene experiments to verify scalability (S3DIS and SemanticKITTI)"; "Supplemented deployment metrics (latency, memory) for more comprehensive experiments"; "Systematic ablation studies quantifying component contributions" (Revisions addressing `Reviewer 7Z2Q`).

- **Presentation:** "Well-organized with readable figures" (`Reviewer vEiz`); "Well-written and easy to follow" (`Reviewer hWC7`); "Good presentation with coherent logic" (`Reviewer vvZF`); "Intuitive and persuasive feature visualization" (`Reviewer hWC7`).

In response to the reviewers’ questions and suggestions, we have made substantial updates to the manuscript, including supplementing strong ANN baselines, expanding large-scale scene experiments, adding ablation studies and deployment metrics, clarifying technical details, and correcting format issues.

These updates are highlighted in blue in the revised PDF. A summary of changes follows:

- ***Main Text Tables:***

    ○ Added comparative data (accuracy, FLOPs, energy consumption) of strong ANN baselines such as PointMLP, PTv3, and PoinTramba, unifying the evaluation protocol (Addressing `vEiz`, `vvZF`).

    ○ Added new Table 4 (S3DIS indoor semantic segmentation) and Table 5 (SemanticKITTI outdoor LiDAR segmentation) to verify large-scale scene adaptability (Addressing `vvZF`, `7Z2Q`, `hWC7`).

    ○ Added new Table 6 reporting training/inference latency and peak memory to improve deployment evaluations (Addressing `hWC7`).

    ○ Modify an ablation table quantifying independent contributions of the SNN framework and hybrid architecture on Table 7, clarifying differences between "Baseline/+SNN/+Hybrid/+Full Model" (Addressing `7Z2Q`).

- ***Appendix Supplements & Clarifications:***

    ○ To address Reviewer `vvZF`’s second response on Mamba’s adaptation to non-causal point clouds, cited existing experimental data and Spike Firing Rate visualization (Figure 3) to demonstrate bidirectional scanning’s effectiveness.

    ○ Added detailed derivation of computational complexity in Appendix I, confirming SLOA’s local attention (O(N)) and SMB’s linear complexity to verify overall O(N) rationality (Addressing `hWC7`).

    ○ Added new Table 17 in Appendix E.7 with energy consumption data for ScanObjectNN variants, clarifying voting as test-time augmentation without affecting inherent model energy consumption (Addressing `vEiz`).

- ***Data & Format Corrections:***

    ○ Regenerated references using ICLR’s official BibTeX template to ensure format compliance (Addressing `vEiz`).

    ○ Optimized Figure 1’s input-output labels, adjusted Figure 4’s layout, and removed duplicate figures to enhance readability (Addressing `7Z2Q`).

- ***Terminology & Architecture Clarifications:***

    ○ Strengthened core motivation description, clarifying SLOA-SMB synergy: SLOA addresses local geometric modeling and permutation invariance, while SMB breaks through global modeling complexity bottlenecks (Addressing `vvZF`).

    ○ Specified SMB’s bidirectional scanning strategy (L-SSM+C-SSM), emphasizing permutation invariance guaranteed by SLOA’s symmetric operations (supported by existing ablation data) (Addressing `hWC7`).

    ○ Explained voting as a standard test-time augmentation (referenced in PointMLP, Mamba3D) to mitigate random inference errors, not a core model component (Addressing `vEiz`).

We look forward to constructive communication with the reviewers to further enhance the quality of our work.

Best regards,

Authors of Submission7340

---

### Meta-Review · Area_Chair_zsz4 · 2025-12-09

**Summary:**

Most reviewers are positive about this submission and find it solid across methodology, evaluation, and presentation. The approach is described as logically and technically sound, with innovative core modules and a coherent co-design that addresses limitations of isolated models. The paper provides comprehensive experiments supporting its claims, showing excellent performance on both classification and segmentation across diverse datasets. Reviewers also consider the manuscript well organized, clearly written, and easy to follow, with readable figures, a coherent narrative, and intuitive feature visualizations.

During the rebuttal phase, the authors provided a very detailed response that strengthened the motivation, expanded experimental coverage, and clarified important technical details. Although one reviewer remains skeptical about the motivation and novelty, the AC finds that the overall contributions and empirical validation outweigh these remaining concerns and therefore recommends acceptance.

**Reviewer Concerns:**

Most reviewers acknowledge the contributions of this work, though they originally requested more comparisons, clarifications, and additional studies. The authors have provided a detailed rebuttal with substantial new experimental data, and most reviewers are satisfied with these responses and lean toward acceptance.

One reviewer, however, remains unconvinced by the motivation for using SNNs over efficient ANN models and feels that the method is largely a combination of existing techniques, with concerns about novelty and motivation not fully resolved. The AC understands these concerns and admits that the work leverages many existing components, adapting them to the SNN setting.

Overall, given the improved experimental support, the generally positive reviewer reception, and the solid technical execution, the AC would think that it meets the standard of ICLR. At the same time, because many substantial revisions and experimental additions (e.g., the paper adds two main experiments on larger-scale point cloud scenes) have been made in the updated manuscript, the AC would also consider it reasonable if the paper goes to another round of review for more thorough evaluations on the revised version.

**Reviewer Scores:**

Reviewer vEiz: 6 (confirmed and maintained)
Reviewer vvZF: 2 (replied but still concerned)
Reviewer 7Z2Q: 8 (no reply)
Reviewer hWC7: 8 (confirmed and maintained)

---

### Decision · Program_Chairs · 2026-01-26

Accept (Poster)